



# Model uncertainties of a storm and their influence on microplastics / sediment transport in the Baltic Sea

Robert Daniel Osinski[1], Kristina Enders[1], Ulf Gräwe[1], Knut Klingbeil[1], and Hagen Radtke[1]

[1]Leibniz Institute for Baltic Sea Research Warnemünde, Seestrasse 15, 18119 Rostock, Germany

**Correspondence:** Robert Daniel Osinski (robert.osinski@io-warnemuende.de)

**Abstract.** Microplastics (MP) are omnipresent in the aquatic environment where they pose a risk to ecosystem health and functioning. Little is, however, known about the concentration and transport patterns of this particulate contaminant. Measurement campaigns remain expensive and assessments of regional MP distributions need to rely on a limited number of samples. The prediction of potential MP sink regions in the sea would thus be beneficial for a better estimation of MP concentration

levels and a better sampling design. Based on a sediment transport model, this study investigates the transport of different MP model particles, PET and PVC particles with sizes of 10 and 330 μm, under storm conditions. A storm event was chosen because extreme wave heights cause intense sediment erosion down to depths unaffected otherwise, and are therefore critical for determining accumulation regions. The calculation of metocean parameters for such extreme weather events is subject to uncertainties. This sensitivity study targets the propagation of uncertainty from the atmospheric conditions to MP erosion and

deposition, on the basis of freely available models and data. We find that atmospheric conditions have a strong impact on the quantity of eroded and deposited material. Thus, even if the settling and resuspension properties of MP were known, a quantitative transport estimation by ocean models would still show considerable uncertainty due to the imperfect knowledge of atmospheric conditions. The uncertainty in the transport depends on the particle size and density, transport of the larger and denser plastic particles only takes place under storm conditions. Less uncertainty exists in the location of erosional and depo-

sitional areas, which seems to be mainly influenced by the bathymetry. We conclude, while quantitative model predictions of sedimentary MP concentrations in marine sediments are hampered by the uncertainty in the wind fields during storms, models can be a valuable tool to select sampling locations for sedimentary MP concentrations to support their empirical quantification.

## 1   Introduction

The presence of MP particles has been proven in a variety of different ecosystems (e.g. Huerta Lwanga et al., 2016; Andrady,

2011). MP constitute potential transport vectors for toxic substances, both substituted chemicals during production and adsorbed environmental pollutants, which can be assimilated by aquatic organisms (Besseling et al., 2019). The littering of the environment with these synthetic particles foreign and incompatible to natural cycles is happening at an unprecedented rate and contributes to the degradation of ecosystem services worldwide (Watkins et al., 2017). The relevance of these particulate pollutants for specific ecosystems cannot, however, be assessed when drivers of their distribution are not understood and their

current stocks remain unknown.





Currently, MP data collection from various environmental compartments is expensive and time consuming, consequential only small data sets are achievable. Here, numerical models known and vigorously applied in sediment transport studies can help to complement sparse measurements.

Plastic denotes a wide range of different polymer types along with different density ranges. Among the most widely produced (PlasticsEurope, 2019) are polyvinylchloride (PVC) with a density of $1275\,\mathrm{kg\,m^{-3}}$ and polyethylene-terephthalate (PET) with $1400\,\mathrm{kg\,m^{-3}}$(Andrady and Neal, 2009), which were used as model particles in the present study.

During cyclone "'Xaver"' in October 2017, mean horizontal bottom water currents exceeded $0.5\,\mathrm{m\,s^{-1}}$ in the bottom water, e.g. in the Arkona Basin (Bunke et al., 2019). It is assumed that significant transport and sorting of larger and denser plastic particles only takes place under such storm conditions. The interest of this study is the identification of potential areas of accumulation of MP particles to support the planning of measurement campaigns by identifying potential areas of interest, because we assume that a stock of high-density plastic particles exists in Baltic Sea sediments.

The idea that storm events determine the relocation of settled MP is supported by old knowledge from the amber hunting community. It is observed that only after strong wave and ocean current activity, amber is beach combed and jewelery hunting becomes profitable. Amber is a naturally occurring polymer with a density range of 1050-1150 $\mathrm{kg\,m^{-3}}$ (similar to MP) and is especially abundant in the Baltic Sea. It was produced a long time ago by the resin of trees which now form a standing stock on the Baltic Sea sea floor. In the laboratory measurements by Shields (Shields, 1936), amber was also taken into account. It was found that the initiation of motion of amber can be described by the Shields curve, comparable to that of sediments.

Chubarenko and Stepanova (2017) compared the transport behaviour of amber with the one of MP and found dimensionless critical bottom shear stresses close to the one represented by the Shields curve. They also found a variation depending on the plastic type and shape. Therefore, the Shields curve is adapted to calculate the critical shear stress.

A sediment transport model is applied in this study to simulate the transport of MP as suspended matter with sizes in the order of sand particles. Certain factors cannot be accounted for, such as plastic type and shape which can influence the critical bottom shear stress (Chubarenko and Stepanova, 2017; Enders et al., 2019) and settling velocity of particles (Khatmullina and Isachenko, 2017). Based on laboratory measurements using MP down to 0.4 $\mathrm{mm}$ in size, Waldschläger and Schüttrumpf (2019) calculated a sinking formula depending on the particle shape. For reasons of simplicity, the standard Stokes formula (Stokes, 1851) for spherical particles is used here.

Although the critical bottom shear stress and the settling velocity are assumed to strongly impact the uncertainty in the transport behaviour, this intial study focuses on a quantification of the metocean uncertainty in the transport behaviour. Our objective is to assess whether relocation of MP particles during a single storm event is quantitatively predictable, or whether it is too sensitive to the meteorological uncertainties to allow for a sufficiently precise model estimation. If this uncertainty is too large, even a precise knowledge of a particle's sinking and erosion properties would not allow for an estimation of its transport.

A well-known method to quantify sensitivity to uncertainties in numerical models is the use of an ensemble approach. Ensemble forecasts are used in operational weather prediction since more than 25 years (Buizza, 2018) and were also successfully applied to different areas like, for example, in aviation (e.g. Osinski and Bouttier, 2018), for the energy sector (e.g. Taylor and Buizza, 2003) or in hydrology (e.g. Pappenberger et al., 2008). An application of ensemble forecasts to quantify the uncertainty





in the morphological impact of storms was proposed by Baart et al. (2011). Osinski et al. (2016) applied a windstorm tracking algorithm onto the operational ensemble forecasts of the European Centre for Medium-Range Weather Forecasts (ECMWF) and demonstrated a strong variation of the track as well as of the damage potential of the different realizations of historical storm events in the ensemble members. This range of uncertainty should also be reflected in the uncertainty in the transport

of suspended matter. An ensemble of 30 members, produced by a mesoscale atmospheric model in non-hydrostatic mode, is applied in the presented study to estimate these uncertainties in the transport behaviour of MP.

Existing studies on the transport of MP in the marine environment are mainly based on a particle tracking approach (e.g. Jalón-Rojas et al., 2019b; Liubartseva et al., 2018). Jalón-Rojas et al. (2019a) showed the importance of applying a 3-d model to estimate MP transports. This is the case in this study. An Eulerian approach was applied in our model, i.e. MP is stored as a

concentration in grid cells and a bottom reservoir.

## 2   Data and Models

For our assessment, we applied a four-step model chain, as illustrated in Figure 1. Firstly, ensemble data based on stochastic perturbations were produced with the atmospheric model WRF-ARW to account for uncertainties in the representation of storm events. Secondly, the atmospheric fields were passed to the wind wave model WAVEWATCH III®. Thirdly, atmospheric

and wave ensemble data were then applied to drive the regional ocean model GETM. Finally, a transport module in GETM simulated the transport of PET and PVC with particle sizes of 10 and 330 µm. The atmospheric model WRF-ARW was applied here to produce an ensemble hindcast of a storm surge event in the Baltic Sea and to provide the necessary forcing fields for the wave and the ocean model. The simulation period covered 1 January 2019 to 4 January 2019 UTC. This includes the storm Alfrida[1] which moved across southern Sweden and especially hit the island of Gotland, where wind speeds of 27.5 $\mathrm{m\,s^{-1}}$

(10 Bft) were reached (The Local, 2019). Storms of this strength occur approximately two to three times per year in the Baltic Sea, but at different locations. WAVEWATCH III® (abbreviated as WWIII) was used to produce ensemble hindcasts of wave parameters based on the WRF-ARW output. GETM was driven by the ensemble hindcasts of the corresponding atmospheric and wave parameters from the unperturbed and perturbed model runs.

### 2.1   The atmospheric model WRF-ARW

The atmospheric mesoscale model WRF-ARW[2] (Skamarock et al., 2019) in version 4.1.1 was used in this study for ensemble hindcasting. A region slightly larger than the Baltic Sea is used with a horizontal resolution of about 0.063° and output was written every five minutes. Vertically, 89 pressure levels until 50 hPa were applied in accordance to levels 2 to 90 in the ERA5 reanalysis (Copernicus Climate Change Service (C3S), 2017). Initial and lateral boundary conditions originated from the ERA5 reanalysis. Osinski and Radtke (2020) tested different ensemble generation strategies with WRF-ARW driven by ERA5 and

compared the outcome with the uncertainty measure provided by the ERA5 reanalysis. As demonstrated in Osinski and Radtke

---

[1]e.g. look into the ECMWF Severe Event Catalogue: https://confluence.ecmwf.int/pages/viewpage.action?pageId=129123779 (last access: 02 April 2020)
[2]https://github.com/wrf-model/WRF/releases (last access: 14 March 2020)



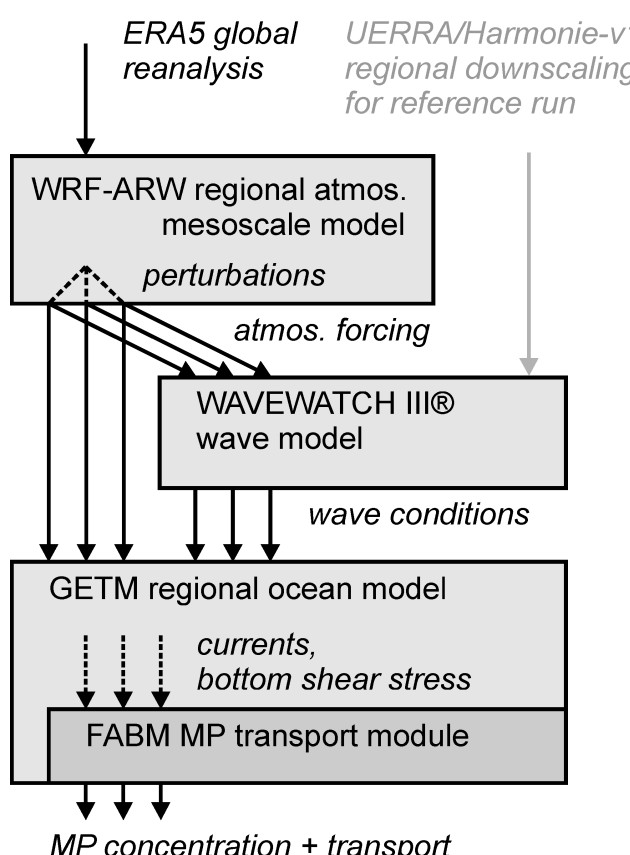

**Figure 1.** Schematic overview of the model chain used in this study

(2020), stochastic perturbations, namely, stochastically perturbed parameterization tendencies (SPPT; Buizza et al., 1999) and stochastic kinetic energy backscatter (SKEB; Shutts, 2005), were used here to produce a small ensemble of 30 members to study the impact of the uncertainty in the atmospheric forcing on the transport patterns, which includes random perturbations of the lateral boundary conditions (Skamarock et al., 2019). Instead of validating the atmospheric data against observations,

5    the wind data were validated indirectly by the wave model output. A visual comparison of the WRF-ARW wind fields against UERRA/HARMONIE-v1 and ERA5 data can be found in Osinski and Radtke (2020).

Osinski and Radtke (2020) compared different ensemble generation methods and proposed to use the ERA5 data from the Ensemble of Data Assimilations as initial conditions to allow for a spread already from the start of the simulation. In contrast, the desired spread needs to develop in the model ensemble in the method chosen here. We chose this method to keep our results

10   comparable to a potential future application in forecast mode. While we ran the model for a storm event in the past, the same could be done for a predicted storm, possibly based on a deterministic forecast product.





## 2.2 The wind wave model WAVEWATCH III®

Wave-induced bottom shear stress is an important driver for the resuspension of bottom sediments and potentially of high-density MP on the seafloor, as investigated in this study. To be able to prescribe wave parameters in high spatial and temporal resolution, the third generation spectral wind wave model WAVEWATCH III v6.07®[3] (Tolman, 1991; The WAVEWATCH III®

Development Group (WW3DG), 2019) was applied in a 3-level one-way nested configuration. The model domain with the highest resolution is based on the same grid as in the GETM model (Gräwe et al., 2019). Dissipation and wind input were based on the formulation of Ardhuin et al. (2010) and the SHOWEX bottom friction scheme after Ardhuin et al. (2003) was applied. For the latter, a map of the D50 sediment grain size was prescribed based on EMODnet[4] data. The wave spectrum was discretized in the same way as in the ERA5 reanalysis with 24 directions starting at 7.5° with a 15° direction increment and 30

frequencies starting at 0.03453 Hz geometrically distributed with a step of 1.1. A setup with 0.1° resolution covering the North Sea and a small part of the eastern Atlantic ocean was used to produce boundary conditions for the Baltic Sea setup at the border with the North Sea. The 0.1° model was nested into a setup for the Atlantic ocean with 0.5° resolution. The GEBCO_2014 Grid in version 20150318[5] was used as bathymetry for the Atlantic and North Sea setups. The Baltic Sea setup had a resolution of one nautical mile with a bathymetry based on the work of Seifert et al. (2001). The 0.5° setup is driven by ERA5 winds and

the ERA5 sea-ice cover fraction. For the 0.1° setup, UERRA/HARMONIE-v1 (Ridal et al., 2017) winds and the ERA5 sea ice cover fraction were used because of their higher spatial resolution. The Baltic Sea setup was driven by two datasets, the UERRA/HARMONIE-v1 wind for a reference simulation and the wind produced with the WRF-ARW wind ensemble for the MP ensemble simulations. Sea ice was taken from the Ostia reanalysis[6]. An obstruction grid based on the GSHHS (Wessel and Smith, 1996) coastline dataset has been generated with the gridgen software[7] to take unresolved orography into account.

Observation data from buoys available from the Copernicus Marine environment monitoring service[8] (CMEMS) were used for validation and calibration. A comparison with station data in Figure 3 shows a good agreement in the significant wave height as well as verification scores over January 2019 (Table 1). The spread in the ensemble is visible at all stations and is expected to provoke differences in the bottom shear stress leading to differences in the resuspension.

## 2.3 The regional ocean model GETM

GETM (General Estuarine Transport Model; Burchard and Bolding, 2002; Hofmeister et al., 2010; Klingbeil and Burchard, 2013) is an ocean model specifically designed for the coastal ocean (see review by Klingbeil et al. (2018)). For the present study, GETM was applied to the Baltic Sea with the model setup of Gräwe et al. (2019), on the same 1 nautical mile grid as the

---

[3]https://github.com/NOAA-EMC/WW3 (last access: 14 March 2020)

[4]http://www.emodnet-geology.eu/ (last access: 14 March 2020)

[5]http://www.gebco.net (last access: 14 March 2020)

[6]http://marine.copernicus.eu/services-portfolio/access-to-products/?option=com_csw&view=details&product_id=SST_GLO_SST_L4_NRT_
OBSERVATIONS_010_001

[7]https://github.com/NOAA-EMC/gridgen (last access: 14 March 2020)

[8]http://marine.copernicus.eu/services-portfolio/access-to-products/?option=com_csw&view=details&product_id=INSITU_BAL_NRT_
OBSERVATIONS_013_032 (last access: 14 March 2020)

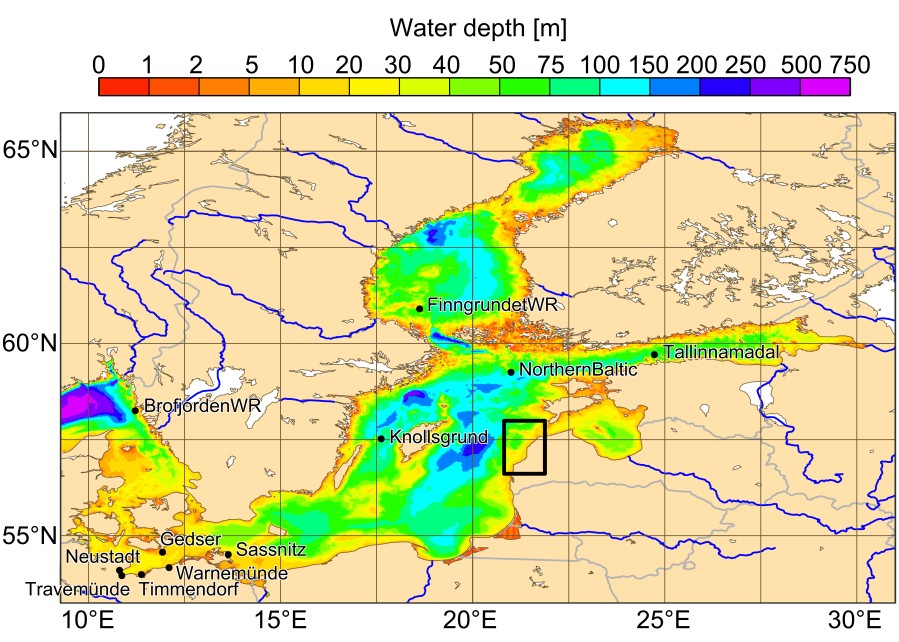

**Figure 2.** Bathymetry [m] of the 1 nautical mile WAVEWATCH III® setup. Black dots show stations for the validation of water level and significant wave height. The black rectangle shows the sub-region for plots of the transport simulation results.

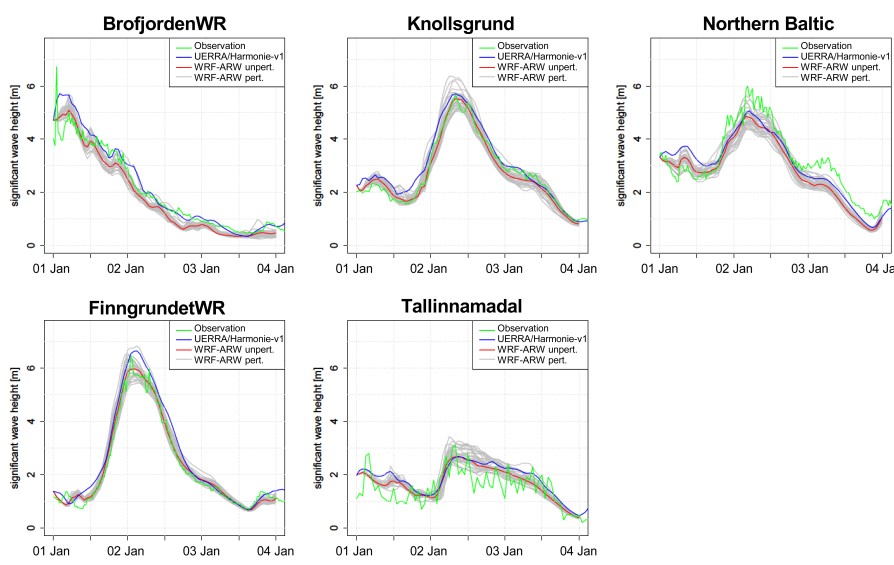

**Figure 3.** Significant wave height at five stations from the 1 nautical mile WAVEWATCH III® model run; wind data from UERRA/HARMONIE-v1, WRF-ARW unperturbed and thirty WRF-ARW members generated with stochastic perturbations.





**Table 1.** Verification scores – root mean square error (RMSE), scatter index (SI, Zambresky, 1989) and correlation (COR) – for significant wave height simulated by WAVEWATCH III® driven by UERRA/HARMONIE-v1 for January 2019

| Station | Bias [m] | RMSE | SI [%] | COR |
|---|---|---|---|---|
| BrofjordenWR | 0.08 | 0.26 | 22.64 | 0.96 |
| Knollsgrund | -0.02 | 0.20 | 15.25 | 0.98 |
| Northern Baltic | -0.11 | 0.29 | 18.33 | 0.96 |
| FinngrundetWR | 0.01 | 0.24 | 18.05 | 0.98 |
| Tallinnamadal | 0.22 | 0.41 | 61.75 | 0.85 |

innermost WAVEWATCH III® nest. The model domain is shown in Figure 2. The original setup was extended by a coupling to FABM (Framework for Aquatic Biogeochemical Models; Bruggeman and Bolding, 2014) to consider sediment and MP. For an accurate 3-d transport of these quantities, GETM provides high-order advection schemes with reduced spurious mixing (Klingbeil et al., 2014), a state-of-the-art second-moment turbulence closure for vertical mixing from GOTM (General Ocean

Turbulence Model; Burchard et al., 1999; Umlauf and Burchard, 2005) and flow-dependent lateral mixing (Smagorinsky, 1963). The accuracy of the model is further increased by adaptive vertical coordinates that guarantee an optimal vertical mesh aligned to the dynamic boundary layers and to the stratified interior (Gräwe et al., 2015). Air-sea fluxes were calculated from the meteorological data provided by the atmospheric model according to the bulk formulas of Kondo (1975). Based on the data provided by the wave model, GETM calculated the mean and maximum combined wave- and current-induced bed stress during

a wave cycle. The latter was used in FABM for the erosion of sediment and MP from the bottom pool (see next section). The initial state of the hydrodynamic model used for this study was obtained by prolonging the simulations from Gräwe et al. (2019) with the atmospheric dataset UERRA/HARMONIE-v1. Further details about open boundary conditions and river discharge can be found in Gräwe et al. (2019).

A detailed validation of the model setup can be found in Gräwe et al. (2019) and Radtke et al. (submitted). For demonstration

purposes, only the spread in sea surface elevation due to the different atmospheric forcing sets is shown here (Figure 4). A verification of the water level at different stations from EMODnet[9] showed a satisfactory performance for both forcing datasets, WRF-ARW and UERRA/HARMONIE-v1. A large spread is also visible in the water level, especially at the peak of the surge.

The ensemble generation in the GETM model in this study is only based on the ensemble hindcasts of the atmospheric and

wave parameters driving the model runs. Brankart et al. (2015) showed that stochastic perturbations in the ocean model are also important for uncertainty estimation. The uncertainty in the ocean currents could therefore be underestimated.

---

[9]https://www.emodnet-physics.eu/ (last access: 14 March 2020)



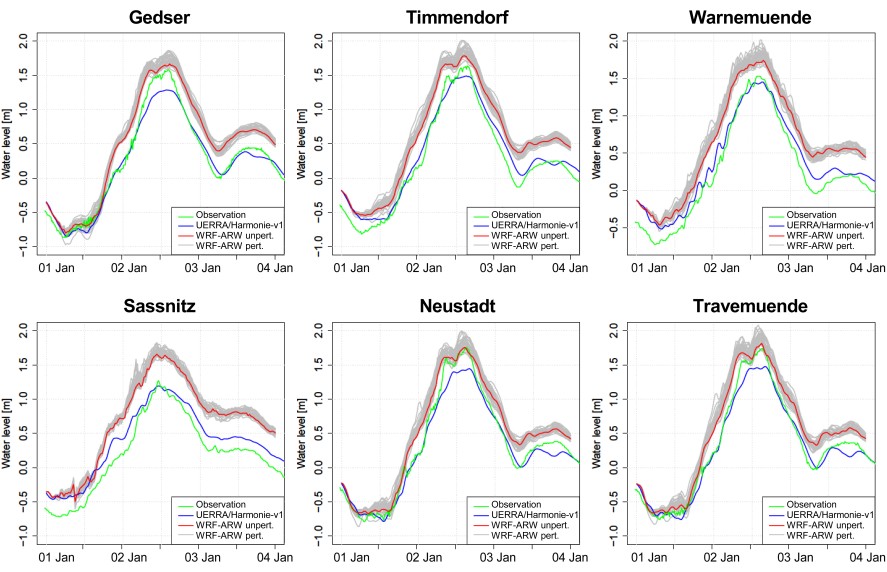

**Figure 4.** Water level at six stations with the 1 nautical mile GETM model; atmospheric data from UERRA/HARMONIE-v1, WRF-ARW unperturbed and thirty WRF-ARW members generated with stochastic perturbations.

## 2.4 Microplastics representation

In GETM and FABM sediment and MP are represented as Eulerian concentration fields. GETM simulated the 3-d transport of the pelagic concentrations, whereas the FABM model calculated the interaction with the corresponding bottom pools due to erosion and deposition and provides settling velocities to GETM. In FABM, a model for non-cohesive sediments (see Sassi

5 et al., 2015) was used to calculate erosion, settling and deposition of both sediment and MP. The different transport was caused by the lower densities of MP, which, however, exceed that of the ambient water, i.e. we only considered sinking particles. This study focuses on model MP of sizes and densities as reported by Stuparu et al. (2015): 10 and 330 µm for both PVC with a density of 1275 $\mathrm{kg\,m^{-3}}$ and PET with 1400 $\mathrm{kg\,m^{-3}}$. To study the impact of density and particle size on the uncertainty in the transport, additional densities of 1100, 1200 and 1300 $\mathrm{kg\,m^{-3}}$ and particle sizes of 200, 250, 300 and 350 µm were tested.

10     The simulations in this study started from homogenous bottom pools of 1 $\mathrm{kg\,m^{-2}}$ as a purely hypothetical reference value and zero suspended material in the water column. Rivers and open boundaries were assumed to not import material into the model domain.





## 3 Results and discussion

### 3.1 MP relocation and its uncertainty

After a 2-days storm surge event, a rearrangement of particles could be observed in the model with some locations dominated by erosion and others by deposition. This can be seen in the change of amount of MP stored in the bottom pool (PET and PVC
with a diameter of 330 µm). To demonstrate the range of uncertainty in the transported amount of MP, two different grid cells in the Gotland basin were selected (Figure 5), 57.69°N 21.35°E (Figure 6a–b) as a net erosion location and 57.66°N 21.32°E (Figure 6c–d) as a net deposition location. Relative to the initial concentration, net erosion varied in the range of 39–72% for PVC and 16–45% for PET. Net accumulation varied between -13–38% for PVC and 22–34% for PET. That is, for PVC in the deposition grid cell (Figure 6c) , in some ensemble members weak erosion is visible while the majority of the ensemble
members show net deposition at this location. For the denser PET, the uncertainty range is smaller than for PVC, implying that its transport is less sensitive to uncertainties in the wind fields and more predictable. Still, the transported amount even in this particle class varies by around a factor of two between realizations, showing that a realistic quantitative estimation of MP transport is impossible in ocean circulation models even if the precise sinking, settling and resuspension properties of the MP particles were perfectly known.

### 3.2 Erosion and deposition areas
Now we consider the spatial patterns where erosion and sedimentation take place. The spatial pattern in four selected ensemble members and the deterministic runs is shown in Figure 7. We chose four members with a considerable spread in the simulated wave height (Fig. 7g). The overall spatial pattern is very similar between the different realizations. The main impact of the metocean uncertainty lies in the amount of the transported material. These findings indicate that the bathymetry has a predom-
inant impact on the region where erosion and deposition take place. For this specific storm surge event and selected region, net deposition took place on the south western sides, net erosion on the north eastern sides of ridges. Model MP of 330 µm in deeper regions, below 50 m, stayed completely unaffected. It is well known that water depth plays a major role for sediment erosion by waves, since deep-water waves (wavelength much shorter than the water depth) show an exponential attenuation in their velocity amplitude with depth (e.g., Kundu and Cohen, 2001). Our findings suggest that this causes stability in spatial
patterns of MP transport against changes in the wind forcing and makes the areas where erosion and deposition take place during a specific storm event predictable.

The uncertainty ranges of the spatial pattern of the model results were further investigated by means of the ensemble statistics composed of the mean, minimum and maximum of each individual grid cell of all ensemble members (Figure 8). The net effect, whether the location was charaterized by deposition or erosion, appeared largely consistent for the entire uncertainty range.
Only few locations showed deviations from this finding where some ensemble members shifted between weak erosional and depositional net effects. The larger extent of the erosional areas was due to more severe representations of the storm event in some ensemble members. Overall, these findings suggest stability in spatial patterns of MP transport against changes in the wind forcing. Areas of erosion and deposition during a specific storm event are predictable.



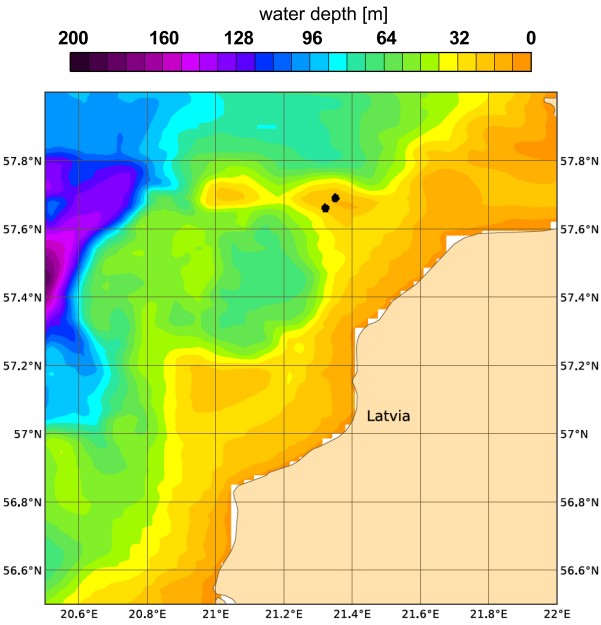

**Figure 5.** Bathymetry [m] of the subregion for which the model results are presented. Black dots indicate the location of two selected grid cells for later reference.

### 3.3 Effect of particle size on transport uncertainty

Next, we investigate the effect of particle size on the uncertainty in the transport, reducing the size of the particles to 10 μm. The small PET particles show a net erosion almost across the whole model domain due to slower resettlement. That is, they are kept in the water column even after 1.5 days after the storm, at the end of the simulation. This partly explains the large difference

between the ensemble minimum and maximum (Figure 9b,c): When sedimentation takes longer, quantitative differences in erosion strength will result in larger transport deviations, since the material can be advected further. This finding is also supported by theory on sediment transport: smaller particles (if unconsolidated) go into suspension under lower shear stress levels and respectively require calmer metocean conditions to deposit. Thus, the uncertainty in MP transport appears to strongly depend on particle diameter and density.

To find out whether this is a systematic effect, the uncertainties in the amount of transported material dependent on the particle properties size and density were investigated in more detail. These relationships were studied based on sensitivity runs with thirty ensemble members for (1) PVC with grain sizes of 200, 250, 300 and 350 μm as well as (2) 330 μm MP of different densities of 1100, 1200, 1300 and 1400 $\mathrm{kg\,m^{-3}}$ (Figure 10a,b). The seafloor concentrations at the end of the model run deviate between the ensemble members. Relative deviations from the ensemble mean were calculated. Figure 10c,d shows

that with decreasing density and/or particle diameter, the relative uncertainty is increasing, with the exception of the 1100 μm MP class showing a smaller uncertainty since it is almost completely resuspended at the chosen location. We conclude, that


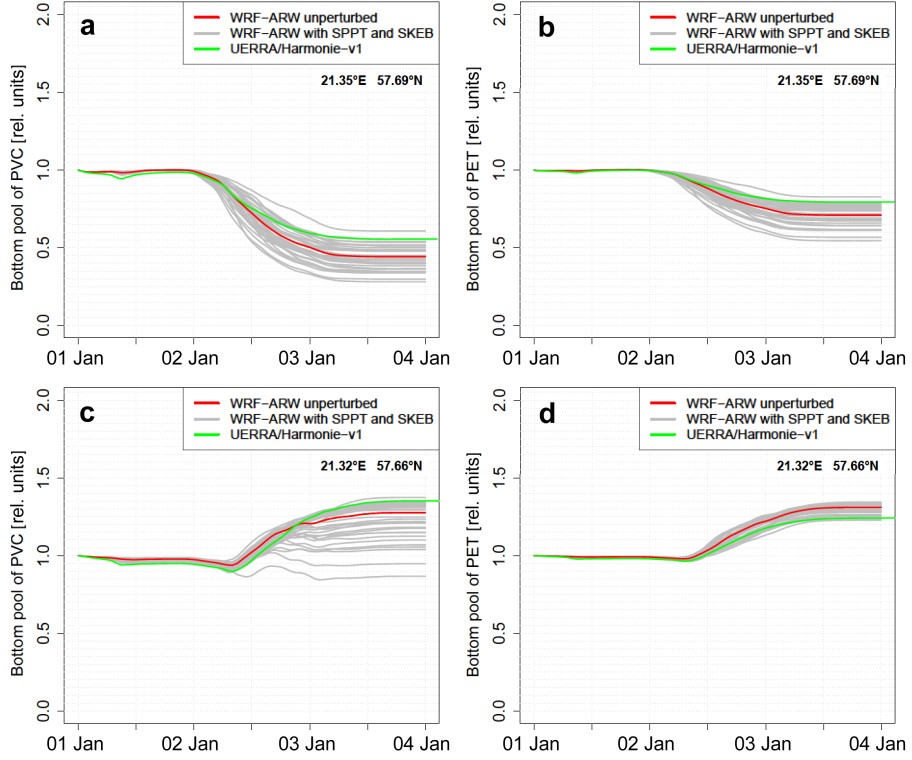

**Figure 6.** Changing bottom concentration of PVC (left panels) and PET (right panels) particles with 330 μm diameter in two grid cells indicated in Figure 5, relative to the initial concentration. The different curves show thirty perturbed runs and one unperturbed run with WRF atmospheric forcing and another simulation with UERRA/HARMONIE-v1 forcing. Panels (a) and (b) show a grid cell predominated by processes of net erosion, whereas (c) and (d) show a cell with net sedimentation.

the uncertainty of the amount of transported material on the seafloor at a specific time depends strongly on the properties of the transported material. The application of an ensemble approach (using more than one model realization to predict transport pathways) is therefore especially important if finer and lighter material shall be represented in future model applications.

### 3.4 Pathways of atmospheric uncertainty propagation

5   In the following, the mechanism by which the atmospheric uncertainty affects the MP transport is identified. In our model, this can be caused (a) by influencing the wave height, which changes the bottom shear stress and therefore MP mobilization or (b) by directly affecting the ocean circulation through e.g. momentum input, thereby influencing both mobilization and transport. We focused on these two major pathways and attempted to distinguish their influence. The possibility of interlinkage by wave-current interaction is neglected in the present model cascade. To estimate the respective uncertainties of MP transport of the two

10  mentioned pathways, an ensemble driven with the wave data from the unperturbed WRF-ARW run with the perturbed WRF-ARW atmospheric forcing and vice-versa with perturbed wave data and unperturbed atmospheric data has been conducted. By



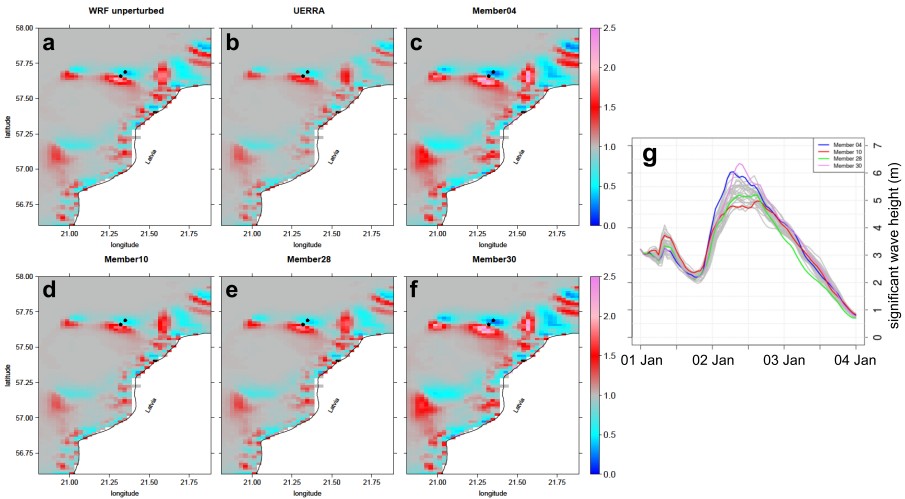

**Figure 7.** Seabed concentration of PVC with 330 µm at 2019-01-03 12UTC, i.e. after the storm surge event in the model, relative to the homogenous initial concentration. Individual panels show the unperturbed WRF run (a), the model driven by UERRA/HARMONIE-v1 (b) and four selected WRF ensemble members (c–f). Dots show the location of the grid cells selected in Figure 6. (g) Timeseries of the significant wave height [m] at the position of the dot in the other figures with net erosion.

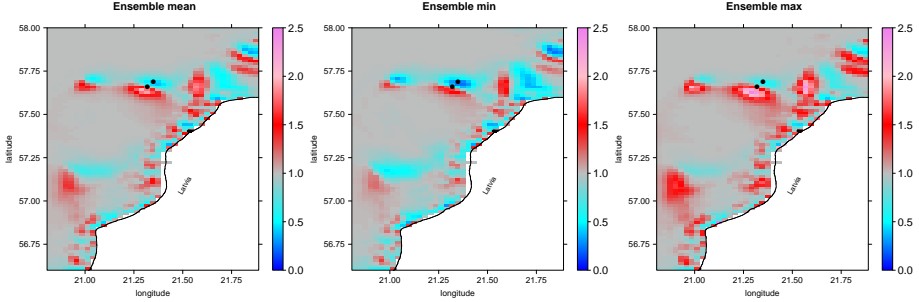

**Figure 8.** Ensemble mean, minimum and maximum of the seabed concentration of PVC with 330 µm at 2019-01-03 12UTC, i.e. after the storm surge event in the model, relative to the homogenous initial concentration. Dots show the location of the grid cells selected in Figure 6.

comparing (Figure 11) the outcome with the original ensemble, where both perturbed atmospheric and wave data were used, it can be seen that the impact of the wave field depends on the properties of the transported material. The lighter or smaller MP, the more important is the impact of the wave uncertainty on the amount of transported material. For denser and larger MP, the uncertainty in the direct effect of atmospheric uncertainty on hydrodynamics is predominant.

5   **3.5   Importance of storms for MP transport**

Higher-density MP of about 300 µm diameter were only transported under severe storm conditions as demonstrated in Figure 12. The continuation of the simulation for the rest of January 2019 caused nearly no further erosion or deposition. This



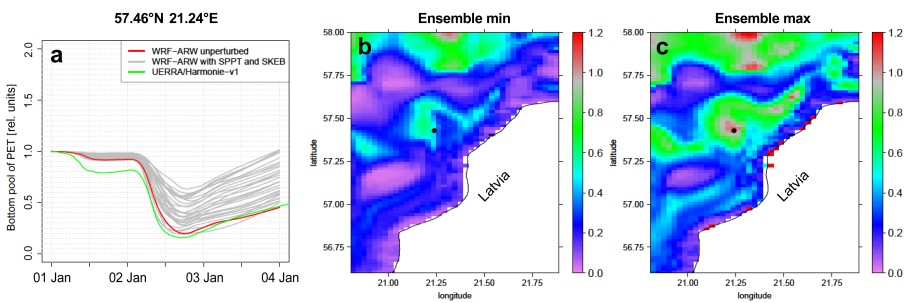

**Figure 9.** (a) Change of the seafloor concentration of PET particles with 10 μm diameter in one selected grid cell in thirty perturbed runs and one unperturbed run with WRF forcing and one run with UERRA/HARMONIE-v1 forcing. (b) Ensemble minimum and (c) ensemble maximum at 2019-01-04 00UTC (at the end of the simulation). All concentrations relative to the homogenous initial concentration. The black dots show the location for the time series plots.

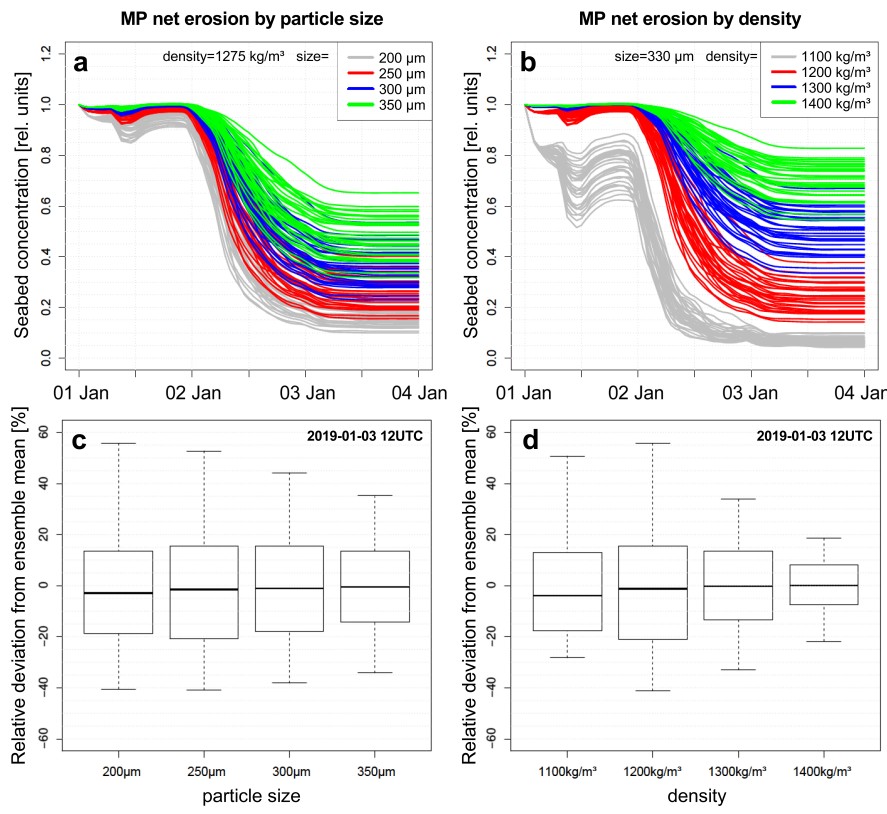

**Figure 10.** Time series of thirty ensemble members at 57.69°N 21.35°E for (a) different MP sizes and (b) different MP densities. (c,d) Box-and-whisker plots show the uncertainty in the concentration of material on the seabed, expressed as a relative deviation of the individual ensemble members from the ensemble mean.

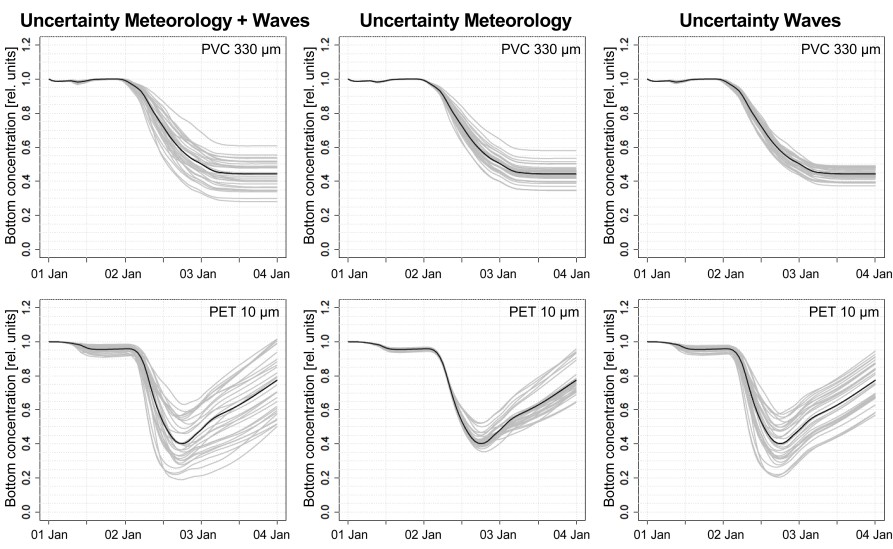

**Figure 11.** Spread of runs with varying atmospheric forcing and/or varying wave forcing, for PVC with 330 μm size (upper panels) and PET with 10 μm size (lower panels). Bottom concentration at 57.69°N 21.35°E (see Figure 9) relative to the initial value.

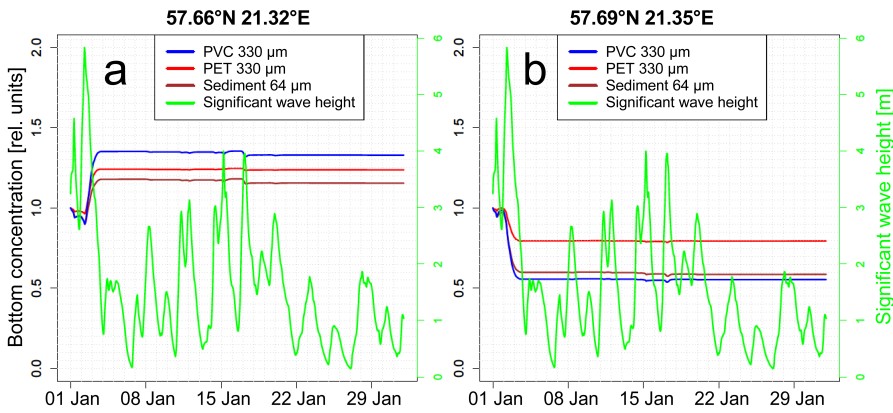

**Figure 12.** Evolution of the amount of PET and PVC with 330 μm and sediment with 64 μm on the sea floor during January 2019, starting from initial amount of $1\,\mathrm{kg\,m^{-2}}$, at two grid cells, a) with net deposition and b) with net erosion.

confirms the assumption of the importance of extreme events for MP transport, which complicates its direct empirical determination. Budget methods will be required to empirically determine quantities of transported MP. That is, transport rates might be more reliably derived from observed amounts before and after storm events than by multiplying abundances of suspended MP with instantaneous volume transports, both of which might show strong temporal variation during extreme weather conditions.





## 3.6 Similarities between MP and sediment transport

The finding that spatial patterns of MP can be reliably predicted by ocean models, while the quantitative estimation of MP was prone to considerable uncertainties shows that additional approaches are required for a more reliable estimation of large-scale MP concentration levels. Here, the recently found MP-sediment proxy postulated by Enders et al. (2019) which is based on

correlations between certain high-density polymer size fractions ($> 1000\,\mathrm{kg\,m^{-3}}$, $> 500\,\mathrm{\mu m}$) and sediment grain size fractions, would be an achievable method. Estimations of MP levels can be based on a relatively small in-situ data set and extrapolated to larger spatial scales by using the MP-sediment correlates. Lower densities of MP (1000 - 1600 $\mathrm{kg\,m^{-3}}$) compared to sediments (quartz: $2650\,\mathrm{kg\,m^{-3}}$) are offset by a larger size. This relationship was explained by comparable threshold bed shear stresses, and thus erosion rates, between these size fractions, which appeared to be the predominant mechanism determining the sorting

in the described study area (Warnow estuary, Baltic Sea, Germany, (Enders et al., 2019)). Although the MP size ranges covered in the present study were below the ones investigated by Enders et al. (2019), it is assumed that similar patterns can be found for smaller size ranges. Indeed, in the present study, after the storm surge event, model PVC of 330 μm co-occurred with sediment grains of 64 μm in size, as apparent by the high correlation coefficient shown in Figure 13. This correlation is found to be largely explained by similar erosion rates (Figure 12b), whereas bottom concentrations predominantly determined by deposition are

also influenced by the settling velocity of particles and thus slightly differ (higher amounts of PVC). It is thus expected that areas largely influenced by the settling of MP show a larger difference in size than described by the current MP-sediment proxy. For instance, larger (and/ or heavier, such as PET) MP particles than 330 μm PVC would be closer to the deposition rate of sediment grains of 64 μm (Figure 12a). Existing maps of sediment substrate type, which typically differentiate between median grain sizes above and below 63 μm (e.g., EMODnet, 2020), may therefore also provide information about MP concentrations

to be expected. However, as this investigation is purely based on our model results with the above-discussed uncertainties, in-situ measurements are inevitable to further research the influences on this MP-sediment proxy.

## 4 Conclusions

A storm surge event in the Baltic Sea in January 2019 has been hindcasted by a four-step model chain. A homogeneous distribution over the entire Baltic Sea was assumed due to a lack of knowledge about the real initial distribution. The model

chain showed a good performance in water level and significant wave height compared to different station data.

The ensemble approach showed a strong variation in the amount of transported MP between ensemble members. This illustrates that quantitative modelling of MP transport during storm events exhibits substantial uncertainty already because the meteorological forcing fields (wind speeds) are imprecisely known. A test with different particle sizes and densities showed a dependence of the uncertainty in the transport on the particle properties.

The spatial distribution pattern where material was eroded or accumulated in the model runs was stable against the atmospheric perturbations. This illustrates the capability of a numerical model to identify regions of interest where seafloor samplings of MP concentrations are promising after the occurrence of a storm.

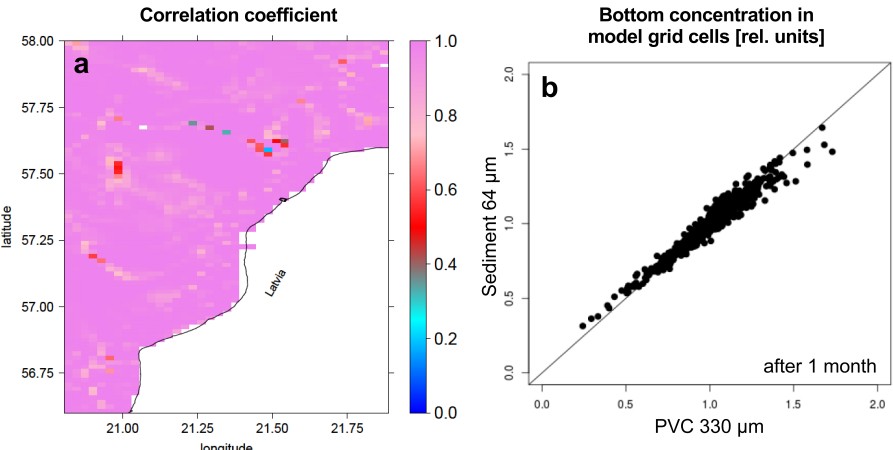

**Figure 13.** (a) Pearson correlation between the time series of bottom concentrations of PVC with 330 μm and sediment with 64 μm for January 2019. (b) Scatter plot of bottom concentrations after the 1-month simulation. Concentrations are given relative to the homogenous initial concentration.

The demonstrated procedure could also be applied in forecast mode, by exchanging the ERA5 reanalysis data used in this study by, for example, the freely availabe GFS forecasts[10]. As a synoptic scale winter storm event is well predictable in the medium-range (3-5 days), this would allow to produce ensemble simulations of MP transport a couple of days in advance to identify sampling regions, as a strategic support tool for measurement campaigns. As strong storm events occur infrequently,

there is a good chance to provide them to sampling campaign planners in time, which means before the next event that could be able to perturb the relocation patterns again. The impact of the uncertainty from the lack of knowledge of settling velocities and critical bottom shear stresses would then have to be taken into account. One idea to reduce the necessary computational resources is a clustering of the atmospheric ensemble data and by driving the rest of the model chain (wave and ocean model) by a reduced set of representative ensemble members.

As the spatial pattern under severe storm conditions is not strongly affected by the uncertainty in the metocean forcing, and transport during moderate conditions can be assumed to be substantially smaller, this study indicates that it would be in principle possible to construct a map of the spatial distribution of high density MP particles in the Baltic Sea using long model runs containing several storm events. The presented study investigates the effect of the uncertainty in the representation of a single storm event. To get a more general picture of erosional and depositional regions in the Baltic Sea, other storm events

with different tracks have also to be taken into account. Also, the spatial pattern and the quantities of MP input, e.g. from river discharge, would need to be known.

The demonstrated ensemble approach can also be useful for other applications like, e.g., in the maritime transport sector. It could help to predict after a strong storm event whether a safe entering of a harbour by big vessels is still possible or whether the morphodynamic changes are so strong that dredging would be necessary.

---

[10]https://www.emc.ncep.noaa.gov/emc/pages/numerical_forecast_systems/gfs.php (last access: 14 March 2020)





*Code and data availability.* The WRF source code is available from https://github.com/wrf-model/WRF/releases, the WAVEWATCH III® from https://github.com/NOAA-EMC/WW3 and the GETM code from https://www.io-warnemuende.de/getm.html. ERA5 and the UERRA/HARMONIE-v1 reanalysis can be retrieved from the Climate data store at https://cds.climate.copernicus.eu.

*Sample availability.* The demonstrated model results can be requested by contacting the corresponding author.

5  *Competing interests.* The authors declare that there is no conflict of interest.

*Acknowledgements.* This study was financed by the Bonus Micropoll project, which has received funding from BONUS (Art 185), funded jointly by the EU and Baltic Sea national funding institutions. K. Klingbeil acknowledges project M5 (Reducing spurious diapycnal mixing in ocean models) of the Collaborative Research Centre TRR 181 "Energy Transfer in Atmosphere and Ocean" (project 274762653) funded by the German Research Foundation (DFG). For the simulations, computing resources at the North German Supercomputing Alliance
10  (HLRN) were consumed. Observational data originate from the E.U. Copernicus Marine Service Information. The simulations in this study were generated using Copernicus Climate Change Service Information (2018/2019). The research and work leading to the UERRA data set used in this study has received funding from the European Union Seventh Framework Programme (FP7/2007-2013) under grant agreement № 607193. We would like to thank the WRF and WAVEWATCH III® developers for providing their models over Github.





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
