# Peer review of "Model uncertainties of a storm and their influence on microplastics / sediment transport in the Baltic Sea"

_Ocean Science, 2020_

## Referee Comment (RC1) · Andrei Bagaev (Referee) · 18 May 2020

**General Comments:**
A study presented in the manuscript investigates the transport of particles (having a given set of properties – density and size) by means of a chain of existing and well established numerical models. They include an atmospheric model, a general ocean circulation model, a spectral wind wave model and two sediment transport models in the Eulerian formulation. From the previous studies (Chubarenko, Enders, Khatmullina) the authors adopted the idea that the behaviour of small polymer particles is somewhat similar to those of natural origin, well-studied previously. This led them

to an assumption that such particles transport reproduced by previously developed models of non-cohesive sediments transport (with modified Shields formulas) may serve as a reliable substitute for microplastics (MPs). Thus the paper addresses a scientific question relevant to the OS scope.

The aim of the research was to obtain an assessment of the sensitivity of the sediments transport model results (erosion, deposition, suspended matter transport) to a stochastic variation in the atmospheric forcing. The authors used an ensemble forecast method, e.g. a simulation with a set of scenarios based on existing reanalysis with introduced stochastic perturbation.

It is concluded that there is a high variability in the amount of transported particles during a storm event. The uncertainty is dependent on the size and density of particles. Meanwhile the space patterns of erosion and deposition areas were stable. The authors promote the use of the chain of models to forecast possible zones of MPs accumulation in order to plan field surveys.

**Specific Comments:**

1. Please specify what makes it possible to consider your model particles as microplastics. It might also be better to separate the description of the experiments from their interpretation and application to MPs transport prediction.

2. The conclusion made in the last sentence in Abstract is poorly linked to the aim of the study and was hard to understand. Please clarify.

3. Introduction, 2nd paragraph: again two poorly linked sentences. It is not clear how the models can complement field measurements.

4. 4th paragraph: too many assumptions made unexpectedly for the reader. Maybe there is a need for more references. New assumptions could be formulated in the Methods section. 'The interest of this study' is not mentioned anywhere in Abstract.

5. Lack of references to existing models. For example: Ballent, A., Pando, S., Purser, A., Juliano, M. F., and Thomsen, L.: Modelled transport of benthic marine microplastic pollution in the Nazaré Canyon, Biogeosciences, 10, 7957–7970,

https://doi.org/10.5194/bg-10-7957-2013, 2013.

Nicole Kowalski, Aurelia M. Reichardt, Joanna J. Waniek Sinking rates of microplastics and potential implications of their alteration by physical, biological, and chemical factors, Marine Pollution Bulletin, Volume 109, Issue 1, 2016, Pages 310-319, ISSN 0025-326X, https://doi.org/10.1016/j.marpolbul.2016.05.064.

A. Bagaev, A. Mizyuk, L. Khatmullina, I. Isachenko, I. Chubarenko, Anthropogenic fibres in the Baltic Sea water column: Field data, laboratory and numerical testing of their motion, Science of The Total Environment, Volumes 599–600, 2017, Pages 560-571, ISSN 0048-9697, https://doi.org/10.1016/j.scitotenv.2017.04.185.

If the transport of the MPs in the marine environment could not be investigated with the existing models, please explain.

6. Why exactly do you prefer to use the Eulerian approach?

7. Both papers KhatmullinaIsachenko and WaldschlägerSchüttrumpf report settling velocities for still fresh water. Please explain the applicability of their results to salt (brackish) turbulent marine water. How exactly do you use those formulas for the settling velocity?

8. You have not mentioned the values of critical sedimentation/resuspension shear stress and settling velocity for your particles. It might be useful for the future studies and the experiments reproduction.

9. It is important to explain why you use 10 and 330 mkm as the size of the particles, which is not common for MPs studies.

10. Page 4: final paragraph - is really hard to understand. Please clarify.

11. Page 9, line 19: 'findings indicate that bathymetry has predominant impact', how exactly do they do this? Is this statement somehow new compared to the results of Enders et al, 2019? I think that Fig. 13 might help you to highlight the new findings.

12. The authors found that with the decrease of MP density and size the ability of models to predict their transport decreases. I think this result is sufficiently supported by the experiments and should be stated more clearly! In fact you showed that small and light MPs (so called nanoplastics) are being driven by waves, while MPs (0.5-5

mm) are affected by hydrodynamics.

13. Page 11, lines 1 and 2 – seems too obvious.

14. Page 14, 'budget methods' – please explain, what do you mean? The whole paragraph looks unclear. 15. Conclusion section – too many repetitions with the Introduction and methods.

16. Important, but somewhat discussionable is the idea regarding possible future application of the chain of models for MPs sink prediction. Your findings are based on the numerical experiments with the spectral wave model and GCM models with 1 nm grid, which might be ok for the sediments, but MPs distributions show high patchiness and probably high mesoscale variability. Which means that your models might require higher spatial resolution in order to be able to determine possible accumulation zones for the samples collection (since in situ samplings of bottom sediments for MPs are usually sparse and low in volume).

**Finally:**

The manuscript represents a good contribution to scientific progress in the microplastics modelling studies, fits the scope of Ocean Science and provides substantial new ideas. It utilizes valid scientific approach and applied models. However the results discussion was somewhat unclear, and the links between some of the expressed ideas seemed not obvious to me. In the Introduction and Methods section too many assumptions are made in order to link sediment transport to the MPs transport. I suggest changing the Title so that 'uncertainties' and 'modelling' look more important than 'microplastics'. The scientific results are presented in a clear, concise, and well-structured way, but I would like to ask to shorten and highlight better the conclusions drawn from those results. Since I have specific comments that may require a major revision of certain paragraphs, I prefer to skip the list of the language evaluation and leave it to the stage after resubmission.
* * *

---

## Referee Comment (RC2) · Florian Pohl (Referee) · 29 May 2020

Osinski et al. numerically simulated the transport and dispersal of high-density plastic particles in the Baltic Sea during a storm event, and demonstrate how atmospheric models can predict microplastic concentrations in seafloor sediments in shallow waters. These predictions could guide seabed sampling campaigns and potentially help to estimate seafloor plastic budgets. This paper, I believe, will be of interest to the readership of Ocean Sciences as it is, as far as I know, the most comprehensive study integrating atmospheric and sediment transport models to predict sediment transport on the seafloor.

[Figure]

**OSD**

A particular storm event ('Alfrida', 1st – 4th Jan 2019) was simulated using freely available and established numerical models. Field measurements of water level changes and wave height at different stations were used to validate the accuracy of the model. A storm generates movement of the water underneath the sea surface, at a depth range and intensity depending on the amplitude of the surface waves. The here presented model predicts by combining atmospheric with sediment transport models, whether the water movement at the seabed might mobilize and suspend sediment. Assuming an initially homogenous distribution of spherical microplastics on the seabed, the model then estimates the transportation and re-distribution of these particles during the storm. This allows to identify areas on the seabed which are enriched or depleted in microplastics relative to the initial concentration. Further the authors test different variations of the models to assess the accuracy of different steps in their model chain.

Despite these positive aspects, I feel, however, that the authors should include a more detailed discussion on the basic role of storms and sea-surface waves in the transport and distribution of clastic sediment and plastics on the seafloor. The assumptions for the modeling are all mentioned in the manuscript, but their implications for the results and conclusions are not discussed sufficiently. I also think that the text could be clarified in places, making the paper more accessible to non-experts on atmospheric modelling. Below I included some comments to the authors.

Yours sincerely, Florian Pohl

**Main comments**

– I missed a discussion on the relevance of storms as a sediment transport mechanism on the seafloor. What about other sediment transport processes such as seafloor currents (e.g. tidal, thermohaline, hyperpycnal flows, river discharges etc.) and sediment gravity flows (e.g. slides or turbidity currents – likely to be triggered by storm events)? To which water depth can a storm event affect the seabed? Typically, the storm-weather wave-base is located at  150 – 200 m, and sediments below this base

are unaffected. Could the authors explain how storms can transport sediment across the seafloor? In the rock record, storm deposits (Hummocky cross-stratification) indicate mainly reworking of sediment on the seafloor, rather than lateral transportation.

– The sediment transport model could be explained clearer. I struggle to understand what this model is doing exactly. How was the bed shear stress calculated and what are the assumptions for these calculations? What type of movement is simulated at the seabed (oscillatory water motion by waves or unidirectional flow)? What are the values of the calculated shear stress and do these make sense when comparing to field and laboratory measurements? I think the outreach of the paper would increase significantly if it becomes clearer to non-experts what this model is doing. In particular as this paper will be of high interest and relevance for readers from other research fields. I cannot evaluate the atmospheric models, as this is not my field of expertise.

– The used criterion for the movement or suspension of sediment is not clear. The Shields curve describes the initiation of movement of sediment on the bed, which means transportation as bedload. There exist additional curves to estimate the threshold for suspension of sediment (e.g. (Bagnold, 1966; van Rijn, 1993; Nino et al., 2003)). Could the authors be more specific which criterion they used and why? Also, the Shields criterion describes the movement of particles under unidirectional flow. How would this translate through to oscillatory water motion, as caused by wave movement?

– Assumptions and limitations of the model should be discussed. The authors specifically state all assumptions and simplifications in their calculations, but I was missing a discussion on how these assumptions (e.g. spherical particles) might affect the results and conclusions.

**Comments made while reading the manuscript:**

Page 1, line 7: Can you mention to which depth these surface waves would reach down the water column?

Page 1, line 13-15: Would this also depend on the ocean depth? Maybe you mean this with 'bathymetry'? I suggest to specifically mention that the ocean depth plays a major role in whether or not particles on the seafloor can be resuspended due to increased surface wave intensity.

Page 2, line 2-3: Could you back this up with a reference? At least in deep-marine sedimentology, sediment transport models still have issues and results often do not match observations.

Page 2, line 8-9: Who assumes that? What about other sediment transport processes such as seafloor currents or sediment gravity flows?

Page 2, line 16-20: Could you please be more specific here. The Shields curve would give you the critical shear stress at which particles would start to move as bedload. Other curves describe the initiation of suspension (e.g. (Bagnold, 1966; van Rijn, 1993; Nino et al., 2003)). Also, this diagram estimated the critical shear stress with a unidirectional flowing current. It is not clear to me how this would translate through to oscillatory water motion, as caused by wave movement.

Page 2, line 20: How have you calculated the shear stress exerted on the seabed due to wave motion of the sea surface?

Page 7, line 3-7: This needs more explanations. These sentences are difficult to understand.

Page 7, line 9: What is the difference between wave and current induced bed (shear) stress? I guess this relates back to my comment on page 2, line 16-20.

Page 7, line 10: Does this mean that the wave induced oscillatory motion of the water at the seafloor is neglected? Looking at ancient storm deposits in the rock record, oscillatory motion appears to be a dominant sedimentary process.

Page 7, line 10-13: It is not clear to me what this means. If this is important, it should be explained. If not, these sentences might be removed from the manuscript.
Page 7, line 15: Sea surface elevation = water level?

Page 8, line 9: Why did you chose these particular grain-size range? What about particles between 10 and 200 $\mu$m?

Page 9, line: 17: Please amend to: Figure 7c-f.

Page 9, line 19-21: Was there a predominant current direction? Could you indicate this direction in figure 7? Could this current explain the pattern of erosion and deposition (i.e. erosion on northeast and deposition on southwest dipping slopes)? Would this pattern change if the direction of the storm surge is different?

Page 9, line 22-24: I think it is very important to state that surface waves can only re-distribute sediments and plastics to a certain water depth. Storms are important for the MP distribution in coastal areas and shallow seas (e.g. large areas of the Baltic Sea), but apparently play a minor role in the distribution of MPs on the seafloor for most parts of the world's oceans (below the storm weather wave base). I would think that MPs are frequently re-mobilized by storms and thus get transported until they are deposited below the storm weather wave base. Here water depth is too deep and plastics cannot longer be re-mobilized by storms. This would suggest that MP concentrations are probably highest just below the storm weather wave base.

Page 9, line 24-26: What about changes in the wind direction?

Page 10, line 5: Why is the color scheme in figure 9b c different compared to figure 7 8? This is confusing and makes a comparison difficult.

Page 10, line 12: Again, why where these grain-size classes chosen? Wouldn't it make more sense to spread the size classes more evenly in-between the two end-members (10 and 300$\mu$m)?

Page 11, line 5-9: I think I finally understood that you model both, oscillatory motion and unidirectional flow. Is this correct (see my comment on page 7, line 10)? How high are the calculated bed shear stresses?

Page 12, line 1-2: Does atmospheric forcing mean the generation of a unidirectional current at the seabed? If yes, what is the current velocity and how did you account for interactions with the bathymetry?

Page 14, line 1-2: What about sediment transport mechanism other than storm induced movements? Tidal currents for example. Although tidal currents are not very strong in the Baltic Sea, they play a significant role in the North Sea. Storms may also trigger sediment gravity flows such as turbidity currents which could transport MPs on the seafloor (Pohl et al., 2020). Also seafloor currents due to thermohaline circulation can transport and re-distribute MPs (Kane et al., 2020). I think other processes should be discussed.

Page 14, line 2: This is an interesting point. Could the authors be more specific on how these budget methods would work?

Page 15, line 12-13: The authors use in their model only spherical particles, although most MPs have more complex shapes (angular and oblate fragments, fibers etc.). I fully understand that the simulation of more realistic shapes would add another level of complexity, or might be even impossible to model as we don't fully understand the hydrodynamics for these complex shapes (Khatmullina and Chubarenko, 2019). However, the authors should mention possible deficiencies of the model due to the assumption of spherical particles. Nevertheless, I think these models are crucial for understanding MP distributions and the assumption of spherical particles is a good starting point.

Page 15, line 15-16: I don't understand this sentence. Size difference in what? MPs? Sediment? Could the authors please rephrase and make this clearer?

Page 15, line 18-20: Only because they have the same settling velocity? I think that this is too simple.

Page 15, line 31-32: This is only valid for shallow waters above the storm weather wave base.

Page 16, line 10-13: This is very interesting! Could these models predict particular microplastic sinks on the seafloor? To which water depth would storms affect the seafloor distribution of MPs?

**Used References**

Bagnold, R. A., 1966, An Approach to the Sediment Transport Problem from General Physics.: USGS Professional Paper 422-1, U.S. Government Printing Office, 42 p., doi:10.1017/S0016756800049074.

Kane, I. A., M. A. Clare, E. Miramontes, R. Wogelius, J. J. Rothwell, P. Garreau, and F. Pohl, 2020, Seafloor microplastic hotspots controlled by deep-sea circulation: Science, v. 5899, no. April, p. eaba5899, doi:10.1126/science.aba5899.

Khatmullina, L., and I. Chubarenko, 2019, Transport of marine microplastic particles : why is it so difficult to predict? v. 305, no. September, p. 293–305.

Nino, Y., F. Lopez, and M. Garcia, 2003, Threshold for particle entrainment into suspension: Sedimentology, v. 50, p. 247–263, doi:10.1046/j.1365-3091.2003.00551.x.

Pohl, F., J. T. Eggenhuisen, I. A. Kane, and M. A. Clare, 2020, Transport and Burial of Microplastics in Deep-Marine Sediments by Turbidity Currents: Environmental Science Technology, v. 54, no. 7, p. 4180–4189, doi:10.1021/acs.est.9b07527.

van Rijn, L. C., 1993, Principles of sediment transport in rivers, estuaries and coastal seas: Amsterdam, Aqua publications. 790 pp., 790 p., doi:10.1002/9781444308785.

---

## Author Response (AR1)

Dear Dr. Bagaev,

Thank you for your review of our manuscript. Please find in the following our responses to your comments. We repeated your comments in bold and you can find our response in italic.

**1. Please specify what makes it possible to consider your model particles as microplastics. It might also be better to separate the description of the experiments from their interpretation and application to MPs transport prediction.**

*Our aim of this study is to investigate in how far the uncertainty in the representation of extreme storm events in metocean data for the Baltic Sea affects the uncertainty in the transport of sediment and MP. For this purpose, we simplified the representation of MP in the model. The idea of using a sediment transport model for transport simulations of MP is motivated by the cited studies. As a simplification, we assume that the plastic particles have a spherical shape and a density defined by the two high density plastic types (PVC and PET). Based on these simplifications, there is additional uncertainty in the transport simulation resulting, for example, from non-optimal settling velocities and critical shear-stresses, but this kind of uncertainty was not to be quantified in this study. We show different kinds of experiments, and some of the experiments are motivated by the outcome of another. For this reason, we decided to keep the description of an experiment and its interpretation closely together to allow the reader to follow this logic.*

**2. The conclusion made in the last sentence in Abstract is poorly linked to the aim of the study and was hard to understand. Please clarify.**

*If forecasting a storm event with a state-of-the-art weather model, the location and intensity of a storm system is affected by uncertainties which originate from uncertainties in the initial conditions, lateral boundary conditions and the model physics. The purpose of this study is to investigate if these uncertainties also affect the location of areas where material during/after the storm event is eroded/deposited, because in the different representations of the storm (ensemble members), its track varies in its position. The study indicates that the uncertainty in the storm representation is affecting the amount of transported material, but the location of erosional and depositional areas keeps nearly constant in the study area (changes only in size because of more or less erosion). This means that the model chain can be used in forecast mode to predict areas where erosion/deposition takes place. This allows for a strategic planning of measurement campaigns, because the model can be used to identify regions in which we should take samples. We will make this clearer.*

**3. Introduction, 2nd paragraph: again two poorly linked sentences. It is not clear how the models can complement field measurements.**

*As explained for the previous comment, the model chain allows for identifying regions in which erosion/deposition should take place. Our aim is not complementing the measurement campaigns, but to have a tool which can be used to identify sample regions beforehand. The proposed model helps to identify regions in which larger amounts of high-density MP is potentially deposited. This allows for a more specific planning of measurement campaigns.*

**4. 4th paragraph: too many assumptions made unexpectedly for the reader. Maybe there is a need for more references. New assumptions could be formulated in the Methods section. The interest of this study is not mentioned anywhere in Abstract.**

*We apply a simplification of a MP transport model to study the impact of the metocean uncertainty on the sediment and MP transport. Aim of this study is to investigate, whether this uncertainty affects the location of erosional and depositional areas. The application of the sediment transport model is motivated by the cited articles. We will make it clearer that the parameters for the MP transport are simplified and better motivate the purpose of the study, a decision support tool for measurement campaign planning.*

**5. Lack of references to existing models. For example: Ballent, A., Pando, S.,Purser, A., Juliano, M. F., and Thomsen, L.: Modelled transport of benthic ma-rine microplastic pollution in the Nazar Canyon, Biogeosciences, 10, 79577970, https://doi.org/10.5194/bg-10-7957-2013, 2013.Nicole Kowalski, Aurelia M. Reichardt, Joanna J. Waniek Sinking rates of microplasticsand potential implications of their alteration by physical, biological, and chemical-factors, Marine Pollution Bulletin, Volume 109, Issue 1, 2016, Pages 310-319, ISSN0025-326X, https://doi.org/10.1016/j.marpolbul.2016.05.064.A. Bagaev, A. Mizyuk, L. Khatmullina, I. Isachenko, I. Chubarenko, Anthropogenicfibres in the Baltic Sea water column: Field data,**

laboratory and numerical testingof their motion, Science of The Total Environment, Volumes 599600, 2017, Pages560-571, ISSN 0048-9697, https://doi.org/10.1016/j.scitotenv.2017.04.185.If the transport of the MPs in the marine environment could not be investigated with the existing models, please explain.

*The studies that we know so far use a deterministic representation of the metocean conditions for the transport simulations, i.e. they calculate MP transport under the assumption that the wind conditions were exactly known. They focus on parameters like the settling velocity, for example. These parameters for the transport model are simplified in our study, instead we use probabilistic metocean data. We mentioned in the conclusions that for a better prediction of the MP transport, we would have to improve the parameters for the MP transport model. The existing studies would also get an additional source of uncertainty if applying probabilistic instead of deterministc metocean data. We will add references to existing models and make the difference and the different focus to existing studies clearer.*

**6. Why exactly do you prefer to use the Eulerian approach?**

*The idea is to apply a sediment transport model, because these models are widely used in coastal engineering for example. The physics described by Eulerian and Lagrangian models is the same, the difference is just the numerical implementation. So when a sufficient spatial resolution / number of particles are used, it shouldn't make a difference which method is applied.*

**7. Both papers Khatmullina Isachenko and Waldschläger Schüttrumpf report settling velocities for still fresh water. Please explain the applicability of their results to salt (brackish) turbulent marine water. How exactly do you use those formulas for the settling velocity?**

*We use the Stokes formula as a simplification for the settling velocity. In an improved version of the model, the settling velocity could be represented by the mentioned articles. For example, we could use an ensemble approach based on different parameters to represent the uncertainty in the settling velocity, or define different fractions of the same plastic particle with different settling velocities based on the distribution of the particle shapes. A combination of the ensemble of metocean conditions with a representation of the uncertainty in the parameters for the MP transport (settling velocity, critical shear stress) would improve a forecast of MP transport processes.*

**8. You have not mentioned the values of critical sedimentation/resuspension shear stress and settling velocity for your particles. It might be useful for the future studies and the experiments reproduction.**

*These parameters do not have constant values since they depend on sea water viscosity. We, however, give example values for $10°C$ water in a new appendix section now.*

**9. It is important to explain why you use 10 and 330 mkm as the size of the particles, which is not common for MPs studies.**

*These diametres are motivated by a study for the North Sea (Stuparu et al., 2015). In this way we have MP particles which correspond to a relatively fine and coarse sediment fraction. We will include this information in the text.*

**10. Page 4: final paragraph - is really hard to understand. Please clarify.**

*The uncertainty of weather forecast originates in uncertainties in the initial conditions, the lateral boundary conditions and the representation of the model physics. For processes which cannot be explicitly resolved by the model resolution, parameterizations are used. We use stochastic perturbations of these parameterizations. The methods applied here are standard methods used at various operational forecast centres. The cited study tested to use initial conditions from an ensemble of data assimilation. In this way, the uncertainty in the initial conditions will lead to differences (spread) between the ensemble members already in the first model time steps. In the presented approach, it needs some time until the stochastic perturbations provoke differences in the members.*

**11. Page 9, line 19: findings indicate that bathymetry has predominant impact, how exactly do they do this? Is this statement somehow new compared to the results of Enders et al, 2019? I think that Fig. 13 might help you to highlight the new findings.**

*The motivation of this article is to investigate the impact of metocean uncertainty on the transport behaviour of MP. The finding in the presented study is related to this uncertainty, which is a result of the uncertainty*

*in the metocean data used to drive the sediment transport model.*

**12. The authors found that with the decrease of MP density and size the ability of models to predict their transport decreases. I think this result is sufficiently supported by the experiments and should be stated more clearly! In fact you showed that small and light MPs (so called nanoplastics) are being driven by waves, while MPs (0.5-5mm) are affected by hydrodynamics.**
*The study focuses on the uncertainty in the MP transport provoked by the uncertainty in the representation of a storm in metocean data. We found a larger uncertainty for smaller and lighter material, which shows that an ensemble approach is getting more important if one is interested in smaller and/or lighter particles. The uncertainty in ocean currents and waves also differs with particle properties. A short-coming of this study is the fact that there are no stochastic perturbations of the model physics of the ocean model. For this reason, the uncertainty in the hydrodynamics might be underestimated.*

**13. Page 11, lines 1 and 2  seems too obvious.**
*Our statements show that if one is interested in the modelling of very light and/or very small material, the uncertainty in the metocean forcing of the transport model becomes more important. We do not know any study taking this kind of uncertainty into account.*

**14. Page 14, budget methods  please explain, what do you mean? The whole paragraph looks unclear.**
*A budget method relates (a) input and (b) output of a quantity to (c) changes in its mass, e.g. inside an area of interest. If two of the three values are known, the third one can be determined. The purpose of our study is a potential support for the planning of measurement campaigns. To be able to create a map of the sea-floor with MP concentrations, a better knowledge of concentrations entering the Baltic Sea is necessary. We assume a homogeneous distribution over the sea-floor. This is sufficient to see where potential erosion and deposition could take place. For a more realistic simulation, knowledge about the amount of material inside the Baltic Sea would be necessary. Then, the model could run for a longer period, and should approximate the distribution on the sea-floor. The error in the approximation will be size- and density dependent.*

**15. Conclusion section  too many repetitions with the Introduction and methods.**
*We will revise the conclusion section and remove repetitions.*

**16. Important, but somewhat discussionable is the idea regarding possible future application of the chain of models for MPs sink prediction. Your findings are based on the numerical experiments with the spectral wave model and GCM models with 1 nm grid, which might be ok for the sediments, but MPs distributions show high patchiness and probably high mesoscale variability. Which means that your models might require higher spatial resolution in order to be able to determine possible accumulation zones for the samples collection (since in situ samplings of bottom sediments for MPs are usually sparse and low in volume).**
*Our interest is a decision support for planning measurement campaigns. This is why we are interested in regions where large amounts of material is potentially deposited after a storm event. We think that for this purpose, the resolutions of the models are sufficient. We are also able to nest specific domains with higher resolutions into the existing models. For the western Baltic Sea, we tested setups with 600 m resolution for the wave and ocean models and 1.4 km for the atmospheric part.*

Dear Dr. Pohl,

Thank you for your review of our manuscript. Please find in the following our responses to your comments. We repeated your comments in bold and you can find our response in italic.

**1 Main comments:**

**I missed a discussion on the relevance of storms as a sediment transport mechanism on the seafloor. What about other sediment transport processes such as seafloor currents (e.g. tidal, thermohaline, hyperpycnal flows, river discharges etc.) and sediment gravity flows (e.g. slides or turbidity currents likely to be triggered by storm events)? To which water depth can a storm event affect the seabed? Typically, the storm-weather wave-base is located at 150 200 m, and sediments below this base are unaffected. Could the authors explain how storms can transport sediment across the seafloor? In the rock record, storm deposits (Hummocky cross-stratification) indicate mainly reworking of sediment on the seafloor, rather than lateral transportation.**
*It is a known issue that during strong storm activity, amber is beach-combed. In this study, we focus on high-density MP particles. We assume for this reason that there should be a comparable behaviour of these MP particles and amber. The cited articles underline this assumption. As there is a source of amber in the Baltic Sea, we assume that there are also locations which accumulate MP. A 3-D ocean model is used here with terrain-following vertical coordinates. It is capable to simulate the mentioned sea-floor currents. River discharges are defined from a climatology, but without MP load, as we are interested in the resuspension of particles from the seabed. A simulation over one month, suggests that only strong wave activity during the storm event produced sufficiently high bed-shear stresses to transport the MP particles. This corresponds to the experiences from the amber hunting community. High shear stresses are neccessary to transport the particles in suspension. The water depth still affected by waves depends on the wave length. The Baltic Sea is relatively shallow with the result that larger parts of the seafloor can be affected by wave activity.*

**The sediment transport model could be explained clearer. I struggle to understand what this model is doing exactly. How wromas the bed shear stress calculated and what are the assumptions for these calculations? What type of movement is simulated at the seabed (oscillatory water motion by waves or unidirectional flow)? What are the values of the calculated shear stress and do these make sense when comparing to field and laboratory measurements? I think the outreach of the paper would increase significantly if it becomes clearer to non-experts what this model is doing. In particular as this paper will be of high interest and relevance for readers from other research fields. I cannot evaluate the atmospheric models, as this is not my field of expertise.**
*We will add an appendix to the manuscript explaining the sediment transport model more in detail. It is a 3-D model, which calculates the concentrations in each model grid cell. An empirical formula for the estimation of the combined shear stress of waves and currents at the sea-floor is used. The initiation of motion is calculated by the Shields curve, settling velocity of the particles is simplified and based on the Stokes formula.*

**The used criterion for the movement or suspension of sediment is not clear. The Shields curve describes the initiation of movement of sediment on the bed, which means transportation as bedload. There exist additional curves to estimate the threshold for suspension of sediment (e.g. (Bagnold, 1966; van Rijn, 1993; Nino et al., 2003)). Could the authors be more specific which criterion they used and why? Also, the Shields criterion describes the movement of particles under unidirectional flow. How would this translate through to oscillatory water motion, as caused by wave movement?**
*An empirical formulation is used to estimate the combined impact of currents and waves in terms of an effective bottom stress. Details including the formulas can now be found in an added appendix section.*

**Assumptions and limitations of the model should be discussed. The authors specifically state all assumptions and simplifications in their calculations, but I was missing a discussion on how these assumptions (e.g. spherical particles) might affect the results and conclusions.**

*The spherical shape of the particle will influence its settling velocity. Waldschläger and Schüttrumpf (2019) discuss the impact of the shape on the settling velocity. There is also an impact of biofilms, which affect the weight of the particles. Our aim was to quantify in how far the uncertainty from the metocean conditions affect the transport. The simulations showed a strong impact on the amount of transported material, but not on the location where erosion and deposition takes place. This finding should persist if adapting the parameters to more realistic ones, affecting the amount of transported material and the specific location for a specific size class. The uncertainty is taking into account by driving the model chain with an ensemble of the atmospheric model. The uncertainty in the parameters like the settling velocity or critical bed shear stress could also be taken into account, by defining several fractions covering the uncertain range of the specific parameter. This is possible as there is no interaction between the fractions.*

**2 Comments made while reading the manuscript:**

**Page 1, line 7: Can you mention to which depth these surface waves would reach down the water column?**
*Interaction of the wave with the seafloor takes places in depth less than half the wave length. The dominant wavelength is between 20 m and 70 m and can reach up to 130 m (Kriauinien, J., Gailiuis, B. and Kovalenkovien, M. 2006. Peculiarities of sea wave propagation in Klaipeda Strait, Lithuania. BALTICA 19: 20-29.).*

**Page 1, line 13-15: Would this also depend on the ocean depth? Maybe you mean this with bathymetry? I suggest to specifically mention that the ocean depth plays a major role in whether or not particles on the seafloor can be resuspended due to increased surface wave intensity.**
*With uncertainty, the uncertainty in the representation of the storm is meant. At a fixed position, the amount of eroded or deposited material is affected differently depending on the particle properties. Ocean depth at this location is important, but also in the vicinity of the location, which influences waves and currents. This is meant by bathymetry.*

**Page 2, line 2-3: Could you back this up with a reference? At least in deep-marine sedimentology, sediment transport models still have issues and results often do not match observations.**
*The sediment transport model is based on the work of Sassi et al. (2015), we add this reference. We assume that for the task of identifying areas of interest for empirical quantification of MP accumulation, uncertainties in the transport models are acceptable.*

**Page 2, line 8-9: Who assumes that? What about other sediment transport processes such as seafloor currents or sediment gravity flows?**
*We add references which stress the importance of extreme events for sediment erosion.*

**Page 2, line 16-20: Could you please be more specific here. The Shields curve would give you the critical shear stress at which particles would start to move as bedload. Other curves describe the initiation of suspension (e.g. (Bagnold, 1966; van Rijn,1993; Nino et al., 2003)). Also, this diagram estimated the critical shear stress with a unidirectional flowing current. It is not clear to me how this would translate through to oscillatory water motion, as caused by wave movement.**
*Details are now given in an added appendix session.*

**Page 2, line 20: How have you calculated the shear stress exerted on the seabed due to wave motion of the sea surface?**
*Details are now given in an added appendix session.*

**Page 7, line 3-7: This needs more explanations. These sentences are difficult to understand.**
*These sentences describe properties of the GETM model which reduce undesired numerical mixing. Numerical mixing leads to an unrealistically high diffusion of transported concentrations, reducing the peak concentrations and overestimating the area in which tracers spread. We add an explanatory sentence stating this.*

**Page 7, line 9: What is the difference between wave and current induced bed (shear) stress? I guess this relates back to my comment on page 2, line 16-20.**
*GETM simulates the ocean currents on a 3-D mesh. The current induced bed shear stress is based on this current. Wave data are externally provided from the simulation done with WAVEWATCHIII. Wave induced bed shear stress is calculated based on theses wave data. Both stresses are added also taking their non-linear interaction into account. Details are now given in the appendix.*

**Page 7, line 10: Does this mean that the wave induced oscillatory motion of the water at the seafloor is neglected? Looking at ancient storm deposits in the rock record, oscillatory motion appears to be a dominant sedimentary process.**
*With the latter, the maximum combined wave- and current-induced bed stress is meant. This is based on an empirical formula as mentioned before. So, oscillatory motion is taken into account.*

**Page 7, line 10-13: It is not clear to me what this means. If this is important, it should be explained. If not, these sentences might be removed from the manuscript.**
*For the regional ocean model in this study, initial conditions and lateral boundary conditions are needed. Starting from initial conditions which do not agree with the meteorological data will cause adjustment effects at the very beginning which may produce unrealistically high currents. This statement says how the conditions are at the beginning of the simulation and what goes in and out at the border of the model domain. This is a necessary information and citation.*

**Page 7, line 15: Sea surface elevation = water level?**
*Correct. In line 16, water level is used.*

**Page 8, line 9: Why did you chose these particular grain-size range? What about particles between 10 and 200$\mu$m?**
*We used the same sizes as in the study for the North Sea from Stuparu et al. (2015). It is not a range, these are two discrete fractions, and it is computationally expensive to add more fractions. Our purpose is a support for measurement campaigns and particles of above 300$\mu$m are easier to sample.*

**Page 9, line: 17: Please amend to: Figure 7c-f.**
*Figure a is the deterministic run (without stochastic perturbations of the model physics in the atmospheric part). Figure b serves as a comparison with a publicly available atmospheric dataset.*

**Page 9, line 19-21: Was there a predominant current direction? Could you indicate this direction in figure 7? Could this current explain the pattern of erosion and deposition (i.e. erosion on northeast and deposition on southwest dipping slopes)? Would this pattern change if the direction of the storm surge is different?**
*Currents in the Baltic Sea are in long-term driven by a thermohaline circulation leading to cyclonic currents, but intermittently can be changed and even reversed by wind. This also controls transport direction of the suspended material and consequently deposition areas. An entirely different storm realization could therefore also change erosion deposition patterns, but we see that this effect plays a minor role in our simulations, i.e. meteorological uncertainty is not that strong. We state the main current direction in the caption of the figure but do not add it e.g. as arrows not to mix it up with the wind direction shown in other figures.*

**Page 9, line 22-24: I think it is very important to state that surface waves can only redistribute sediments and plastics to a certain water depth. Storms are important for the MP distribution in coastal areas and shallow seas (e.g. large areas of the Baltic Sea), but apparently play a minor role in the distribution of MPs on the seafloor for most parts of the worlds oceans (below the storm weather wave base). I would think that MPs are frequently remobilized by storms and thus get transported until they are deposited below the storm weather wave base. Here water depth is too deep and plastics cannot longer be remobilized by storms. This would suggest that MP concentrations are probably highest just below the storm weather wave base.**
*We agree with this speculation but we could not demonstrate this in this model study. As it is known from amber, there must be a stock on the seafloor affected under storm conditions. Our assumption is that there is a comparable behaviour with MP. This assumption is based on other scientific studies, and the identification of potential deposition areas with the model can help to support measurement campaigns whose outcome could*

*validate the model runs.*

**Page 9, line 24-26: What about changes in the wind direction?**
*They are taken into account by the ensemble approach. The stochastic perturbations of the model physics provoke slightly different developments of the storms in the different members, not only in intensity, also in the track of the storm. This was one of the principal ideas of this study, to see if this variability in the location of the storm affects also the location where erosion and deposition appears.*

**Page 10, line 5: Why is the color scheme in figure 9b c different compared to figure 7 8? This is confusing and makes a comparison difficult.**
*Figure 9 is for the 10 µm fraction and figures 7 and 8 for the 330 µm fraction. The range of values is different (0 to 1.2) for figure 9 and (0 to 2.5) for figure 7 and 8.*

**Page 10, line 12: Again, why where these grain-size classes chosen? Wouldnt it make more sense to spread the size classes more evenly in between the two end-members (10 and 300µm)?**
*Based on the fractions as in Stuparu et al. (2015). The model can simulate only the transport of discrete size fractions. For one measurement method used in the project, 300 µm was the lower limit for the sampling. With 330 µm, we are 10% above of this lower limit.*

**Page 11, line 5-9: I think I finally understood that you model both, oscillatory motion and unidirectional flow. Is this correct (see my comment on page 7, line 10)? How high are the calculated bed shear stresses ?**
*These vary between zero and 0.090 N m$^{-2}$, strongly depending on water depth.*

**Page 12, line 1-2: Does atmospheric forcing mean the generation of a unidirectional current at the seabed? If yes, what is the current velocity and how did you account for interactions with the bathymetry?**
*The 3-D regional ocean model with 1 n.m. resolution applied in this study simulates the ocean currents close to the seafloor. A 3-D regional ocean model (GETM) is used, which has terrain following vertical coordinates. It calculates the U and V components of the current at each model timestep for each grid cell. The model is driven by the atmospheric data, but also includes river discharges and is driven at the open boundary with the North Sea by lateral boundary conditions of a North Sea ocean model.*

**Page 14, line 1-2: What about sediment transport mechanism other than storm induced movements? Tidal currents for example. Although tidal currents are not very strong in the Baltic Sea, they play a significant role in the North Sea. Storms may also trigger sediment gravity flows such as turbidity currents which could transport MPs on the seafloor (Pohl et al., 2020). Also seafloor currents due to thermohaline circulation can transport and redistribute MPs (Kane et al., 2020). I think other processes should be discussed.**
*As figure 12 suggests, strong wave activity plays the pre-dominant role for erosion and though for the transport in suspension. Our aim was not to quantify which are the pre-dominant processes leading to the transport of MP. We wanted to investigate in how far the uncertainty in a weather forecast would affect the transport behaviour of sediment and MP. Tidal currents play a role in the Danish Straits only but the interior of the Baltic Sea is non-tidal. Turbidity currents cannot be represented in our model since the concentration of suspended matter has no influence on seawater density in the model. Thermohaline circulation, on the other hand, is fully taken into account. We add the missing processes to the discussion of the study's limitations.*

**Page 14, line 2: This is an interesting point. Could the authors be more specific on how these budget methods would work?**
*The same question was asked by Reviewer 1 so we give the same answer. A budget method relates (a) input and (b) output of a quantity to (c) changes in its mass, e.g. inside an area of interest. If two of the three values are known, the third one can be determined. The purpose of our study is a potential support for the planning of measurement campaigns. To be able to create a map of the sea-floor with MP concentrations, a better knowledge of concentrations entering the Baltic Sea is necessary. We assume a homogeneous distribution over the sea-floor. This is sufficient to see where potential erosion and deposition could take place. For a more realistic simulation, knowledge about the amount of material inside the Baltic Sea would be necessary. Then, the model could run for a longer period, and should approximate the distribution on the sea-floor. The*

*error in the approximation will be size- and density dependent.*

**Page 15, line 12-13: The authors use in their model only spherical particles, although most MPs have more complex shapes (angular and oblate fragments, fibers etc.). I fully understand that the simulation of more realistic shapes would add another level of complexity, or might be even impossible to model as we dont fully understand the hydrodynamics for these complex shapes (Khatmullina and Chubarenko, 2019). However, the authors should mention possible deficiencies of the model due to the assumption of spherical particles. Nevertheless, I think these models are crucial for understanding MP distributions and the assumption of spherical particles is a good starting point.**
*Our aim was to study the effect of the uncertainty in the representation of storms on the transport of MP particles. This kind of uncertainty is, as far as we know, neglected in other studies. The simplifications of the model have of course impacts on the transport behaviour, as unknown particle shapes will add even more uncertainty.*

**Page 15, line 15-16: I dont understand this sentence. Size difference in what? MPs? Sediment? Could the authors please rephrase and make this clearer?**
*Will be rephrased.*

**Page 15, line 18-20: Only because they have the same settling velocity? I think that this is too simple.**
*The correlations in figure 13 shows the connection between a sediment particle and a ligther but larger MP particle. Enders et al. (2019) showed such a relation based on measurements.*

**Page 15, line 31-32: This is only valid for shallow waters above the storm weather wave base.**
*Yes, but the Baltic Sea is relatively shallow.*

**Page 16, line 10-13: This is very interesting! Could these models predict particular microplastic sinks on the seafloor? To which water depth would storms affect the seafloor distribution of MPs?**
*Yes, such models should be applicable for this task. The model can predict potential sinks of MP, under the condition that material is available. We assumed a homogeneous distribution of MP over the entire Baltic Sea and we do not have river loads or beach accumulation (as a sink) taken into account. The model predicts areas which are sensitive to a potential deposition. The water depth affected by waves is half of the wavelength, which goes up to 130m in the Baltic Sea, but will strongly dependent on wind direction and especially fetch.*

[revised manuscript text omitted]

Sources of uncertainty in atmospheric model predictions originate from the initial conditions, in case of a regional model

10   also from lateral boundary conditions and from the model physics. Osinski and Radtke (2020) compared different ensemble generation methods and proposed to use the ERA5 data from the Ensemble of Data Assimilations as initial conditions to allow for a spread already from the start of the simulation. The initial conditions in the presented study are based on the high resolution ERA5 reanalysis and the model approach includes perturbations of the model physics and the lateral boundary conditions. In contrast, the desired spread needs to develop in the model ensemble in the method chosen here. We chose this

15   method to keep our results comparable to a potential future application in forecast mode. While we ran the model for a storm event in the past, the same could be done for a predicted storm, possibly based on a deterministic forecast product.

**2.2   The wind wave model WAVEWATCH III[®]**

Wave-induced bottom shear stress is an important driver for the resuspension of bottom sediments and potentially of high-density MP on the seafloor, as investigated in this study. To be able to prescribe wave parameters in high spatial and temporal

20   resolution, the third generation spectral wind wave model WAVEWATCH III v6.07[®3] (Tolman, 1991; The WAVEWATCH III[®] Development Group (WW3DG), 2019) was applied in a 3-level one-way nested configuration. The model domain with the highest resolution is based on the same grid as in the GETM model (Gräwe et al., 2019). Dissipation and wind input were based on the formulation of Ardhuin et al. (2010) and the SHOWEX bottom friction scheme after Ardhuin et al. (2003) was applied. For the latter, a map of the D50 sediment grain size was prescribed based on EMODnet[4] data. The wave spectrum was

25   discretized in the same way as in the ERA5 reanalysis with 24 directions starting at 7.5° with a 15° direction increment and 30 frequencies starting at 0.03453 Hz geometrically distributed with a step of 1.1. A setup with 0.1° resolution covering the North Sea and a small part of the eastern Atlantic ocean was used to produce boundary conditions for the Baltic Sea setup at the border with the North Sea. The 0.1° model was nested into a setup for the Atlantic ocean with 0.5° resolution. The GEBCO_2014 Grid in version 20150318[5] was used as bathymetry for the Atlantic and North Sea setups. The Baltic Sea setup had a resolution

30   of one nautical mile with a bathymetry based on the work of Seifert et al. (2001). The 0.5° setup is driven by ERA5 winds and the ERA5 sea-ice cover fraction. For the 0.1° setup, UERRA/HARMONIE-v1 (Ridal et al., 2017) winds and the ERA5 sea
* * *
[3]https://github.com/NOAA-EMC/WW3 (last access: 14 March 2020)

[4]http://www.emodnet-geology.eu/ (last access: 14 March 2020)

[5]http://www.gebco.net (last access: 14 March 2020)

[Figure]

**Figure 2.** Bathymetry [m] of the 1 nautical mile WAVEWATCH III® setup. Black dots show stations for the validation of water level and significant wave height. The black rectangle shows the sub-region for plots of the transport simulation results.

ice cover fraction were used because of their higher spatial resolution. The Baltic Sea setup was driven by two datasets, the UERRA/HARMONIE-v1 wind for a reference simulation and the wind produced with the WRF-ARW wind ensemble for the MP ensemble simulations. Sea ice was taken from the Ostia reanalysis[6]. An obstruction grid based on the GSHHS (Wessel and Smith, 1996) coastline dataset has been generated with the gridgen software[7] to take unresolved orography into account.

5    Observation data from buoys available from the Copernicus Marine environment monitoring service[8] (CMEMS) were used for validation and calibration. A comparison with station data in Figure 3 shows a good agreement in the significant wave height as well as verification scores over January 2019 (Table 1). The spread in the ensemble is visible at all stations and is expected to provoke differences in the bottom shear stress leading to differences in the resuspension.

Waves affect the seafloor until a water depth of about half the wave length. The dominant wavelength in the Baltic Sea is

10  between 20 m and 70 m and can reach up to 130 m (Kriaučiūnienė et al., 1961).
* * *
[6]http://marine.copernicus.eu/services-portfolio/access-to-products/?option=com_csw&view=details&product_id=SST_GLO_SST_L4_NRT_
OBSERVATIONS_010_001

[7]https://github.com/NOAA-EMC/gridgen (last access: 14 March 2020)

[revised manuscript text omitted]

**4   Conclusions**

15 A storm surge event in the Baltic Sea in January 2019 has been hindcasted by a four-step  probabilistic model chain started from an homogeneous initial MP distribution. The model validation showed a good performance in water level and significant wave height compared to different station data.

 A strong variation in the amount of transported MP between ensemble members  illustrates that quantitative modelling of MP transport during storm events exhibits substantial uncertainty already because  of uncertainties in meteorological forcing fields (e.g. wind speeds). A test with different particle sizes and densities showed a dependence of the uncertainty in the transport on the particle properties. The impact of the metocean uncertainty on sediment and MP transport increases with decreasing particle density and/or size.

The spatial distribution pattern where material was eroded or accumulated in the model runs was stable against the atmospheric perturbations, illustrating the capability of a numerical model to identify regions of interest where seafloor samplings of MP concentrations are promising.

The demonstrated procedure could also be applied in forecast mode, by exchanging the ERA5 reanalysis data used in this study by, for example, the freely availabe GFS forecasts[10]. As a synoptic scale winter storm event is well predictable in the medium-range (3-5 days), this would allow to produce ensemble simulations of MP transport a couple of days in advance to identify sampling regions, as a strategic support tool for measurement campaigns.  The impact of the uncertainty from the lack of knowledge of settling velocities and critical bottom shear stresses would then have to be taken into account. One idea to reduce the necessary computational resources is a clustering of the atmospheric ensemble data and by driving the rest of the model chain (wave and ocean model) by a reduced set of representative ensemble members.

As  a consequence of the insensitive of the location of erosional and depositional areas to the uncertainty in the metocean forcing  and a substantially smaller transport during moderate conditions, this study indicates that it would be in principle possible to construct a map of the spatial distribution of high density MP particles in the Baltic Sea using long model runs containing several storm events.  Differences between storm events might be larger than the uncertainty in  a single event. To get a more general picture of erosional and depositional regions in the Baltic Sea, other storm events with different tracks have also to be taken into account.

The demonstrated ensemble approach can also be useful for other applications like, e.g., in the maritime transport sector. It could help to predict after a strong storm event whether a safe entering of a harbour by big vessels is still possible or whether the morphodynamic changes are so strong that dredging would be necessary.

*Code and data availability.* The WRF source code is available from https://github.com/wrf-model/WRF/releases, the WAVEWATCH III[®] from https://github.com/NOAA-EMC/WW3 and the GETM code from https://www.io-warnemuende.de/getm.html. ERA5 and the UERRA/HARMONIE-v1 reanalysis can be retrieved from the Climate data store at https://cds.climate.copernicus.eu.
* * *
[10]https://www.emc.ncep.noaa.gov/emc/pages/numerical_forecast_systems/gfs.php (last access: 14 March 2020)

*Sample availability.* The demonstrated model results can be requested by contacting the corresponding author.

**Appendix A: Mathematical description of the particle sinking and erosion model**

Sinking velocity of the particles is initially calculated by the Stokes formula,

$$w_{Stokes} \;\; = \;\; \frac{gD^2}{18\nu} \frac{\rho_p - \rho_w}{\rho_w}, \tag{A1}$$

where $g$ is the gravitational acceleration, $D$ is the particle diameter, $\nu$ is the kinematic viscosity of water, and $\rho_p$ and $\rho_w$ are
the densities of the particle and the water. To correct for larger particles whose sinking velocity would be overestimated by the
Stokes formula, a Newtonian correction is applied by an iterative algorithm:

- A Reynolds number is calculated as $Re = 0.64 w_{sink} D / \nu$.

- A relative drag coefficient is derived from this Reynolds number as $C_D = 18.5 / Re^{0.6}$ following Perry and Chilton as
  cited by Khalaf (2009).

- The updated velocity is calculated as $w_{sink} = \sqrt{\frac{4gD}{3C_D} \frac{\rho_p - \rho_w}{\rho_w}}$

which can be understood as a weighted geometric mean between the two velocities $w_{Stokes}$ and $\nu/D$. This correction makes
large particles sink slower than the Stokes formula suggests. We, however, erroneously applied the correction also to the small
particles where it resulted in an undesired upward correction. This has no effect on particle erosion but accelerates redeposition,
which may even lead to an underestimation of the influence of meteorological uncertainty for the small particles in our study.

Erosion takes place when the actual shear stress exceeds the critical shear stress. To determine the critical shear stress, we
follow the Shields curve in its version which was corrected by Soulsby (1997). First, we calculate the dimensionless particle
diameter $D_*$, which relates the particle diameter $D$ to a viscosity-determined length scale, following Rijn (1984):

$$D_* \;\; = \;\; \sqrt{3} \frac{g}{\nu^2} \frac{\rho_p - \rho_w}{\rho_w} D, \tag{A2}$$

where $\nu$ is the kinematic viscosity of water, $\rho_p$ is the particle density and $\rho_w$ is the water density. Then we calculate the critical
shields parameter for non-cohesive grains, $\theta_{cr}$ (also dimensionless), following Soulsby (1997) as cited by Ziervogel and Bohling (2003),

$$\theta_{cr} \;\; = \;\; \frac{0.3}{1 + 1.2 D_*} + 0.055 * \left(1 - e^{-0.02 D_*}\right). \tag{A3}$$

The critical shear stress can then be calculated as

$$\tau_{cr} \;\; = \;\; gD(\rho_p - \rho_w)\theta_{cr}. \tag{A4}$$

| diameter | density | sinking velocity | critical shear stress |
|---|---|---|---|
| ($\mu$m) | (kg m$^{-3}$) | (mm s$^{-1}$) | (N m$^{-2}$) |
| 10 | 1275 | 0.15 | 0.006210895 |
| 330 | 1275 | 8.14 | 0.045142586 |
| 10 | 1400 | 0.20 | 0.009277999 |
| 330 | 1400 | 10.98 | 0.062337737 |

**Table A1.** Sinking velocities and critical shear stress in the model at 10°C.

The actual shear stress is calculated from the wave-induced and the current-induced shear stress, $\tau_w$ and $\tau_c$. The current-induced shear stress itself, however, is also modified by the wave field, as it changes the bottom drag coefficient according to the DATA2 formula given by Soulsby (1997),

$$\tau_m \quad \bar{\equiv} \quad \left(1 + 1.2\left(\frac{\tau_w}{\tau_c + \tau_w}\right)^{3.2}\right)\tau_c \,, \tag{A5}$$

where $\tau_c$ is the shear stress induced by the current in the absence of waves. Both of them are combined depending on the angle $\alpha$ between currents and waves,

$$\tau^2 \quad \bar{\equiv} \quad \tau_w^2 + \tau_m^2 + 2\tau_w\tau_m \cos(\alpha) \,. \tag{A6}$$

If the actual shear stress exceeds the critical one, the deposited material gets resuspended with first-order kinetics, i.e. proportional to its mass in the sediment pool.

The actual values for sinking velocities and critical stresses depend on temperature since it influences sea water viscosity. Values for 10°C are presented in Table A1.

*Competing interests.* The authors declare that there is no conflict of interest.

*Acknowledgements.* This study was financed by the Bonus Micropoll project, which has received funding from BONUS (Art 185), funded jointly by the EU and Baltic Sea national funding institutions. K. Klingbeil acknowledges project M5 (Reducing spurious diapycnal mixing in ocean models) of the Collaborative Research Centre TRR 181 "Energy Transfer in Atmosphere and Ocean" (project 274762653) funded by the German Research Foundation (DFG). For the simulations, computing resources at the North German Supercomputing Alliance (HLRN) were consumed. Observational data originate from the E.U. Copernicus Marine Service Information. The simulations in this study were generated using Copernicus Climate Change Service Information (2018/2019). The research and work leading to the UERRA data set used in this study has received funding from the European Union Seventh Framework Programme (FP7/2007-2013) under grant agreement № 607193. We would like to thank the WRF and WAVEWATCH III® developers for providing their models over Github.